# In-depth analysis of transcriptomes in ovarian cortical follicles from children and adults reveals interfollicular heterogeneity

Ilmatar Rooda [1,2,3] ✉, Jasmin Hassan [1,2], Jie Hao [1,2,4], Magdalena Wagner [1,2], Elisabeth Moussaud-Lamodière [1,2], Kersti Jääger [5,6], Marjut Otala [7], Katri Knuus [7], Cecilia Lindskog [8], Kiriaki Papaikonomou [2,9], Sebastian Gidlöf[1,2,9], Cecilia Langenskiöld [10], Hartmut Vogt [11], Per Frisk[12], Johan Malmros[13,14], Timo Tuuri [7], Andres Salumets [1,2,6,15], Kirsi Jahnukainen [16,17], Agne Velthut-Meikas [3,18] & Pauliina Damdimopoulou [1,2,18] ✉

The ovarian cortical reserve of follicles is vital for fertility. Some medical treatments are toxic to follicles, leading to premature ovarian insufficiency. Ovarian tissue cryopreservation is an established method to preserve fertility in adults and even applied in prepuberty despite unproven efficacy. Here, we analyze transcriptomes of 120 cortical follicles from children and adults for detailed comparison. We discover heterogeneity with two main types of follicles in both age groups: one with expected oocyte-granulosa profiles and another with predicted role in signaling. Transcriptional changes during growth to the secondary stage are similar overall in children and adults, but variations related to extracellular matrix, theca cells, and miRNA profiles are found. Notably, cyclophosphamide dose correlates with interferon signaling in child follicles. Additionally, morphology alone is insufficient for follicle categorization suggesting a need for additional markers. Marker genes for early follicle activation are determined. These findings will help refine follicular classification and fertility preservation techniques across critical ages.

The ovarian follicles that form during fetal development are pivotal for female fertility. A reserve of ~1 million primordial follicles is established when proliferating oogonia are enclosed by granulosa cells and they enter and arrest in the first meiotic cell division in fetal ovaries. No new follicles form after birth; hence, this reserve dictates a woman's reproductive lifespan[1]. Primordial follicles may stay dormant for decades, undergo atresia, or activate to grow. In women, folliculogenesis spans about a year and encompasses two main stages: gonadotropin-independent and -dependent growth[2,3]. During the gonadotropin-independent growth, the activated primordial follicles enlarge via oocyte growth and granulosa cell proliferation and start recruiting theca cells from the adjacent stroma. Around the secondary stage, when the

oocyte is surrounded by at least two cuboidal granulosa cell layers, the theca cells become histologically detectable[3]. Subsequently, the follicle initially becomes responsive, and thereafter dependent on the pituitary gonadotropins FSH and LH that stimulate further growth and steroidogenesis until ovulation[4]. Most follicles degenerate during the early gonadotropin-independent growth. While mTOR and PI3K signaling pathways have been implicated in follicle growth activation[5], the precise factors triggering the activation still remain unknown. MicroRNAs (miRNAs) are short non-coding RNAs expressed in ovarian somatic cells, oocytes, and follicular fluid. While miRNAs have been extensively studied in large pre-ovulatory follicles, there is limited knowledge about their expression and role in human pre-antral follicles[6].

Both child and adult ovaries harbor follicles at all stages of gonadotropin-independent growth[3,7]. Importantly, ovarian cortical tissue with its follicles can be cryopreserved without loss of function, which is the basis of fertility preservation by ovarian tissue cryopreservation (OTC). OTC is an established medical routine for adult women facing gonadotoxic treatments with a high risk of infertility as a side effect, such as chemo- or radiotherapy for diverse cancers and hematopoietic stem cell transplantation[8]. After recovery from the disease, the stored tissue can be auto-transplanted to restore endocrine function and fertility. For adults, this method has proven to be efficient with more than 200 reported live births so far[9]. In contrast, successful reports on live births from tissue collected before puberty still remain scant. The youngest successful auto-transplantation and puberty induction subjects were 9–10 years old by OTC[10,11], and three live births have been documented from tissue cryopreserved between ages 9–14[12–14]. Even without verified efficacy in younger children, OTC is occasionally offered even to newborns[15]. Given that pediatric cancer patients are often gravely impacted by their disease, the risk-benefit ratio must clearly favor the patient for them to undergo additional surgeries like ovarian tissue collection. Therefore, it is important to investigate whether ovarian tissue collected in childhood has the same inherent properties as adult tissue, and if it can be expected to function upon transplantation.

### Table 1 | Patient characteristics for follicle sequencing main study

| Patient | Group | Indication for surgery | Treatment | Number of analyzed follicles |
|---|---|---|---|---|
| Patient 1 | Adult | GRP | Androgen | 16 |
| Patient 2 | Adult | GRP | Androgen | 19 |
| Patient 3 | Adult | GRP | Androgen | 15 |
| Patient 4 | Adult | GRP | Androgen | 10 |
| Patient 5 | Child | FP | Chemotherapy (CED: 3000 mg/m²) | 8 |
| Patient 6 | Child | FP | Chemotherapy (CED: 8328 mg/m²) | 16 |
| Patient 7 | Child | FP | None | 21 |
| Patient 8 | Child | FP | Chemotherapy (CED: 6000 mg/m²) | 9 |
| Patient 9 | Child | FP | Chemotherapy (CED: 5952 mg/m²) | 6 |

*GRP* gender reassignment patient, *FP* fertility preservation, *CED* cyclophosphamide equivalent dose.

Child ovaries, compared to adult counterparts, are under-studied. However, the few available reports imply that age influences cortical follicles. For instance, child ovarian samples behave differently in tissue culture with a lower degree of follicle activation and growth compared to pubertal and adult tissue[16]. This may be linked to the larger proportion of abnormal follicles in children or the prepubertal state of the ovary[16]. Moreover, extracellular matrix (ECM) composition and biophysical properties of ovaries change throughout life, being different in children and adults[17,18]. While multiple studies have explored cell types and follicle dynamics in adult human ovaries[19–23], such studies are still lacking in children.

Here, we mapped molecular signatures in child and adult ovarian cortical follicles by transcriptomic profiling of individual viable follicles. While our findings suggest largely comparable molecular signatures across child and adult follicles, we also discovered numerous differences that could underpin the known differences in follicle biology before and after puberty. We also show that morphology alone is not sufficient to classify follicles, urging the inclusion of molecular profiles in follicle staging. Finally, we discover the significant impacts of chemotherapy on gene expression in child follicles. Collectively, our results hold significance in refining basic ovarian biology, fertility preservation techniques, and strategies to protect the ovarian reserve against gonadotoxic treatments.

## Results
### Study setup
Ovarian tissue from adult gender reassignment patients (GRP) (*n* = 4, age 19–29 years) and children undergoing fertility preservation (*n* = 5, age 1–11 years) were collected and cryopreserved (Table 1). Post-thawing, the cortical pieces were enzymatically and mechanically digested, and then stained using neutral red for viability to enable manual collection of healthy follicles (Fig. 1). Each follicle was individually photographed, lysed, and frozen at −80 °C until RNA library preparation (Fig. 1). The lysate was thawed and divided into a parallel analysis of the RNA content, including both long non-coding and protein-coding RNAs containing a polyA-tail (polyA-RNA) and small non-coding RNAs (miRNA), to explore genome-wide gene expression patterns.

### Two groups of follicles are present in the ovarian cortex
After data quality control, 109 follicles remained in the polyA-RNA analysis. Surprisingly, principal component analysis (PCA) revealed a strong PCA 1 that explained 68% of the variance in the dataset (Fig. 2A). When Uniform Manifold Approximation and Projection for Dimension Reduction (UMAP) was applied, a clear separation of the follicles into

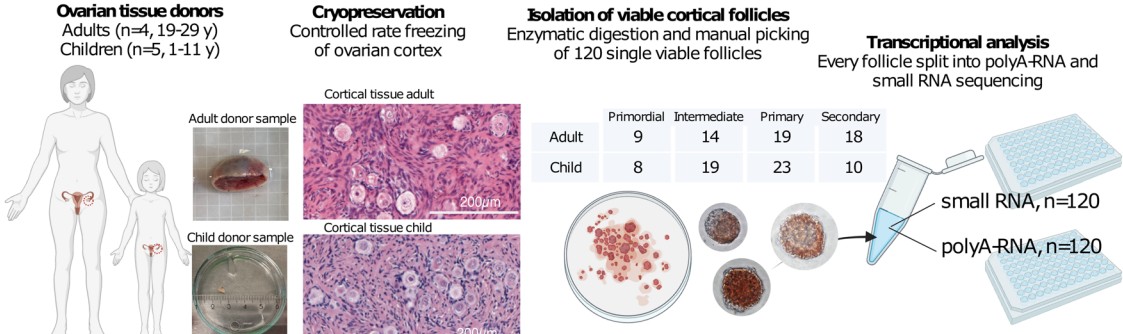

**Fig. 1 | Experimental setup.** Human ovarian tissue was collected from adults undergoing gender reassignment surgery and from children undergoing fertility preservation. Cryopreserved cortical tissue samples were thawed and subsequently dissociated for the manual isolation of cortical follicles under a stereomicroscope. A total of 120 viable follicles, as confirmed by neutral red staining and regular morphology, were chosen for single follicle analysis. Each follicle was individually lysed, and the cell lysate was divided into polyA-RNA sequencing following the Smart-seq2 protocol, and short RNA sequencing using the QIAseq small RNA kit. Created with BioRender.com released under a Creative Commons Attribution-NonCommercial-NoDerivs 4.0 International license.

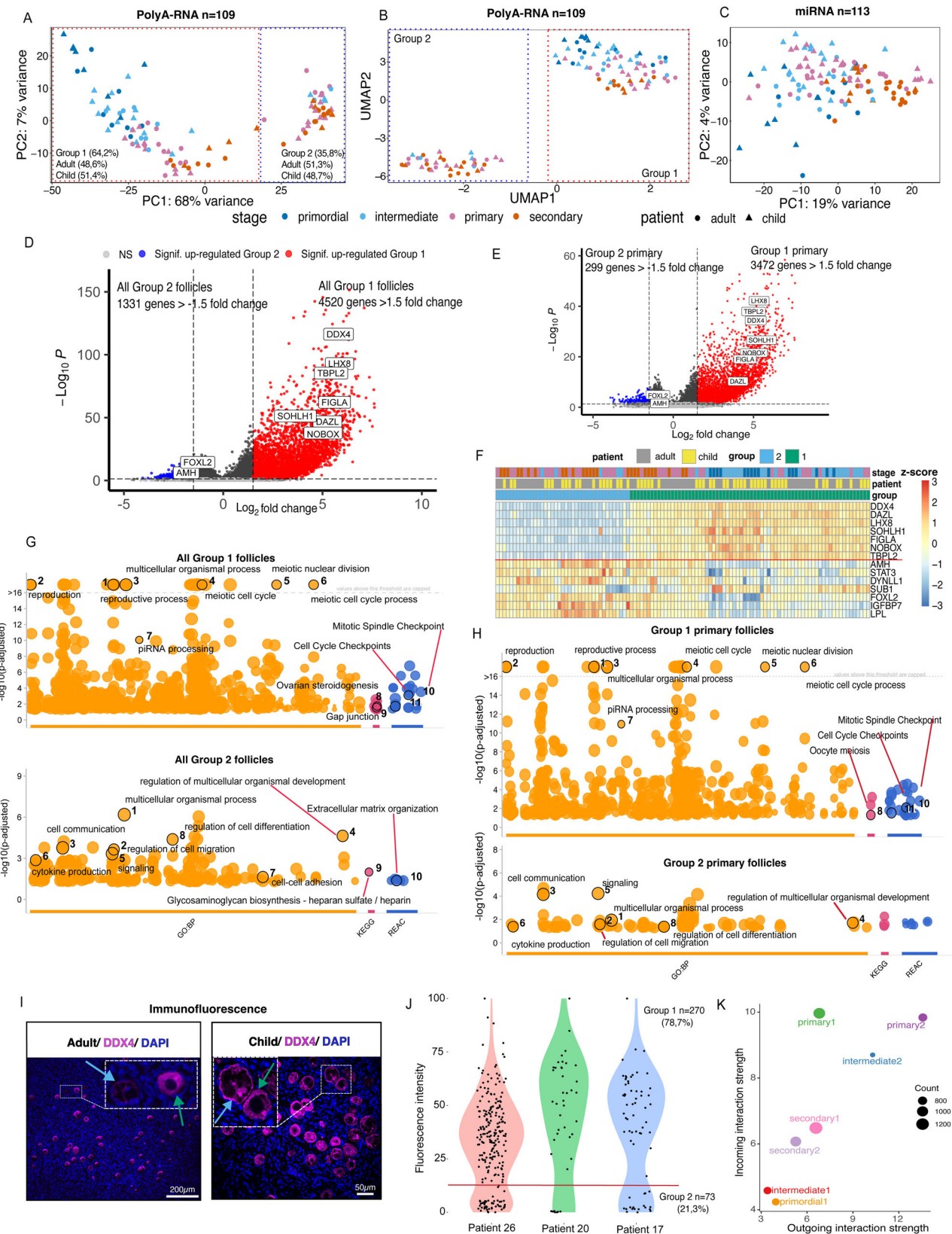

two distinct groups was found (Fig. 2B). The first group (Group 1, *n* = 70, 64% of all follicles) contained all primordial follicles and some of the growing intermediary, primary and secondary follicles, while the second group (Group 2, *n* = 39, 36% of all follicles) contained only growing follicles. The grouping was not affected by the age of the donor, as follicles from children and adults were present in both

clusters, with 34 child follicles in Group 1, and 20 child follicles in Group 2 (Fig. 2A). The grouping was not based on the library count size either, since no significant difference was detected (Supplementary Fig. 1A, B). We considered technical reasons stemming from the splitting of each follicle into two libraries (Smart-seq2 and small RNA-seq) as a possible cause. Therefore, we carried out qPCR analysis of selected

**Fig. 2 | Human ovarian cortex contains two distinct populations of follicles.** 120 follicles of normal morphology isolated from children and adults were analyzed via polyA-RNA and miRNA. After quality control, 109 and 113 follicles remained in the analysis, respectively. **A** Principal component analysis (PCA) and **B** uniform manifold approximation and projection (UMAP) visualization showed two follicle groups based on polyA-RNA expression, with child and adult follicles equally present. Group 1 included all developmental stages, while Group 2 had growing follicles only. **C** PCA based on miRNA expression did not group the follicles but aligned PC1 with developmental stages from primordial to secondary follicles. **D** Volcano plot showed a marked upregulation of genes in Group 1 follicles compared to Group 2, Wald test, adjusted using Benjamini–Hochberg (BH) method. **E** Similar expression profiles were found when specific follicle growth stages were compared; here, primary follicles of Group 1 vs Group 2. **F** Differentially expressed genes (DEGs) between Group 1 and Group 2 follicles included recognized oocyte (above red line) and granulosa (under red line) cell markers. Group 1 exhibited an upregulation of traditional oocyte markers, whereas Group 2 had upregulation of typical granulosa cell markers. **G** The top

enriched Gene Ontologies (GO), KEGG, and Reactome pathways in Group 1 follicles suggested enrichment in reproductive processes, meiotic cell cycle, and steroidogenesis. Group 2 displayed enrichment in cell communication, developmental processes, and extracellular matrix organization, Fisher's one-tailed test, adjusted using BH method. **H** When analyses were restricted to primary follicles only, similar results were obtained. **I** Immunofluorescence staining of adult and child ovarian cortical tissues revealed follicles with varying DDX4 oocyte marker expression (high expression, green arrows; low expression, blue arrows). **J** Quantification of the immunofluorescent signal from 343 follicles across two adults and one child presented a bimodal distribution, suggesting the presence of two follicle groups based on their DDX4 expression levels (red line). **K** Scatter plot of follicle-to-follicle communications in a 2D space. The dot size reflects the number of inferred links associated with each follicle group. Group 2 primary and intermediary follicles were predominant signal senders, while Group 1 primary follicles were the main signal receivers. NS- non-significant. Source data are provided as a Source Data file.

genes in both halves of split follicles from a different set of patients ($n = 2$, GRP patients, $n = 20$ follicles). We found that the results correlated significantly (Supplementary Fig. 1C and Supplementary Data 1A) suggesting that the splitting yields two homogeneous samples. We also sequenced an additional independent validation set comprising 46 follicles isolated from an independent set of patients ($n = 4$, three GRP and one child fertility preservation patient), without splitting them (Supplementary Data 1B). These follicles were also separated into two groups by PCA and UMAP (Supplementary Fig. 1E, F). Finally, we inspected the general appearance of the follicles based on the microscope images, but morphology did not reveal any notable differences between the two groups (Supplementary Fig. 2A). Interestingly, the follicles did not cluster into groups based on their miRNA content, but rather were distributed along PC1 based on morphologically determined developmental stage (Fig. 2C).

Gene expression differed markedly between the two groups, with 12,964 differentially expressed genes (DEGs) identified at FDR < 0.05. Group 1 follicles were characterized by significant upregulation of genes, with prominent oocyte markers such as DEAD-Box Helicase 4 (*DDX4*), Deleted In Azoospermia Like (*DAZL*), and Folliculogenesis Specific BHLH Transcription Factor (*FIGLA*) among the top DEGs (Fig. 2D). In contrast, the top DEGs in Group 2 included notable granulosa cell markers such as Anti-Mullerian Hormone (*AMH*) and Forkhead Box L2 (*FOXL2*) (Fig. 2D). Since follicle stages were unevenly distributed between Group 1 and Group 2, we considered the possibility that the DE analysis results might be skewed. However, when we conducted stage-specific comparisons between Group 1 and Group 2 follicles, we obtained similar results. This suggests that the grouping is not driven by specific follicle growth stages (see primary follicle comparison in Fig. 2E and secondary follicle comparison in Supplementary Fig. 1D). A heatmap based on DEGs representing typical oocyte and granulosa cell markers also distinctly clustered the follicles into two groups: Group 1 with elevated oocyte markers and Group 2 with high granulosa cell markers (Fig. 2F). We considered that Group 2 follicles might be missing their oocyte due to some technical reason; however, several meiosis-specific genes were expressed in both follicle groups (Supplementary Fig. 2B) and the microscope images showed the presence of an oocyte (Supplementary Fig. 2A). In addition, the independent validation set follicles ($n = 4$ patients, $n = 46$ follicles) confirmed the same trends (Supplementary Fig. 1E–H).

Pathway enrichment and gene ontology (GO) analyses for upregulated DEGs in Group 1 (FDR < 0.05, fold change > 1.5) highlighted the enrichment of diverse processes associated with meiosis, cell cycle, steroidogenesis, and reproduction. Enrichment results for all Group 1 follicles are displayed in Fig. 2G, for Group 1 primary follicles in Fig. 2H, for Group 1 secondary follicles in Supplementary Fig. 1D, and for the Group 1 follicles of the independent validation set follicles in

Supplementary Fig. 1I. In contrast, the upregulated DEGs in Group 2 were enriched in pathways related to communication, signaling, development, and ECM. The results for all Group 2 follicles are displayed in Fig. 2G, for Group 2 primary follicles in Fig. 2H, and for the independent validation set Group 2 follicles in Supplementary Fig. 1I. All enriched pathways, GOs, and DEGs related to Group 1 and Group 2 comparisons are further listed in Supplementary Data 2. Importantly, neither group showed enrichment in terms related to cell death, suggesting that the grouping was not related to follicle viability. Given that the follicle selection was based on a viability marker (neutral red stain) and morphology, this was expected. As such, based on their transcriptomic profiles, follicles in both groups seem viable and active.

To rule out the possibility that the distinction between Group 1 and Group 2 follicles arose from the isolation process, we performed immunofluorescent staining on ovarian tissue sections from an independent sample set, comprising of GRPs ($n = 2$), Cesarean section (c-sec) patients ($n = 2$), and child fertility preservation patients ($n = 1$) (Supplementary Data 1C). We targeted the oocyte marker DDX4, one of the top DEGs, in both child and adult tissues containing follicles (Fig. 2I). Quantification of the staining intensity across 343 follicles revealed a bimodal distribution, which approximately divided the follicles into two groups (Fig. 2J, red line). Furthermore, RNA in situ hybridization experiments validated the presence of follicles exhibiting varying levels of *DDX4* and *AMH* transcripts within the ovarian cortex both in children and adults (Supplementary Fig. 2C). Moreover, immunostaining for key oocyte transcription factors, FIGLA and LHX8, further validated follicular heterogeneity, as certain follicles exhibited very low or absent expression compared to others (Supplementary Fig. 2D).

Given that Group 2 follicles displayed elevated granulosa cell markers as well as multiple significantly enriched gene sets related to cell signaling, we studied signaling activities between the follicles using the CellChat tool and databases[24]. Interestingly, we found that the most potent outgoing signals came from Group 2 primary and intermediary follicles, whereas the strongest incoming signals were detected in Group 1 primary follicles (Fig. 2K). Outgoing signals from Group 2 follicles to Group 1 follicles included some expected follicle growth factors such as KIT Proto-Oncogene, Receptor Tyrosine Kinase (*KIT*), and *AMH* (Supplementary Fig. 3A). However, the top secreted signals across all Group 2 follicles were midkine (*MK*) and semaphorin-3 (*SEMA-3*) (Supplementary Fig. 3A). Midkine is a retinoic acid responsive growth factor that stimulates cell growth, migration, and angiogenesis. It has previously been suggested as a novel ovarian reserve marker[25,26]. Semaphorin-3 is a group of chemokines involved in cell migration and angiogenesis that have not previously been linked to folliculogenesis. Although Group 1 follicles also exhibited the capability to secrete a range of growth factors, including both previously identified ones and

factors not linked to folliculogenesis before, the predicted signaling strength was notably lower compared to Group 2 follicles (Fig. 2K and Supplementary Fig. 3B). To assess the robustness of follicle-to-follicle communication, we compared three normalization methods (Cell-Chat, Deseq2, and EdgeR). All three methods yielded comparable results, indicating that Group 2 exhibited a higher outgoing interaction strength compared to Group 1 (Supplementary Fig. 3C–E).

In summary, our results suggest that the ovarian cortex in both children and adults may house viable follicles of normal morphology that separate into two distinct types based on their transcriptomic profiles. While the functions of these potential new follicle types remain uncertain, our findings suggest that only Group 1 follicles might progress normally, given that they express essential oocyte markers. Conversely, these essential markers are significantly reduced in Group 2 follicles, which, however, express typical granulosa cell-derived growth factors and show high communicational activity based on bioinformatic analysis, suggesting a role in local paracrine regulations.

### Adult and child ovarian follicle transcriptional programs

We next investigated the temporal changes in gene expression during folliculogenesis in children and adults. We excluded Group 2 follicles from this analysis for two reasons: (1) because of their low expression of key oocyte transcription factors, and (2) because this group did not contain any primordial follicles. The PCA of Group 1 follicles ($n = 70$, Supplementary Data 3A) showed the predicted sequential progression from the primordial to the secondary follicle stage in children and adults (Fig. 3A). Gene expression changes during follicle development revealed numerous DEGs (polyA-RNA, FDR < 0.05) between the follicular stages (Fig. 3B). Most DEGs were found during the primordial-to-primary transition in both age groups (Fig. 3B), which is similar to previous studies in mouse and human follicles[23,27]. Comparison of DEGs marking the transcriptional changes from primordial to secondary follicles in children and adults revealed a total of 6849 unique DEGs (FDR < 0.05). Only 1469 of these were shared between adults and children, while 3480 were statistically significant only in adults and 1900 only in children (Fig. 3C, Supplementary Data 4A). The observed differential expression of genes in the dataset may be subject to some degree of uncertainty, for instance, due to inter-patient variability, limited sample size, and unequal number of follicles between different comparisons. However, when down-sampling to the lowest follicle number ($n = 6$ in adults, $n = 4$ in children), we observed a substantial overlap in results across all down-sampled sets compared to all follicle sets, indicating we detected robust changes in follicle development (Supplementary Fig. 4C).

To study the overall changes in gene expression during follicle development, we subjected the child and adult DEGs to pattern analysis (Supplementary Fig. 4A, B, Supplementary Data 4B). DEGs were scaled and subjected to hierarchical clustering to group them into patterns of similar expression. Interestingly, the main gene expression patterns were consistent between children and adults (Fig. 3D). In both categories (adult and child), the expression levels of most genes gradually increased as the follicles progressed from the primordial to the secondary stage. Conversely, approximately one-third of the DEGs followed a decreasing trend (Fig. 3D). The child and adult patterns closely mirrored each other, even when the Venn diagram only showed a 21% overlap between the two age group DEGs (Fig. 3D). These results suggest a default gene expression blueprint for ovarian follicle development during the gonadotropin-independent phases in both children and adults.

To elucidate the functions of genes exhibiting consistent upward or downward trends during follicle growth, we pooled the child and adult DEGs with similar patterns (either consistently rising or consistently declining), and evaluated their enrichment into biological processes and pathways. The genes upregulated during follicle growth (pattern 2 in adults, pattern 1 in children) were linked to ECM

organization, biosynthesis, development, and morphogenesis (Fig. 3E and Supplementary Data 5A), whereas the genes downregulated (pattern 6 in adults, pattern 3 in children) were associated with cell cycle, meiosis I, chromosomal segregation, and cytoskeletal organization (Fig. 3F and Supplementary Data 5B). Although enrichment of cell cycle among downregulated genes may seem counterintuitive, cell cycle arrest and negative regulation of cell proliferation characterize human oocytes deriving from primordial and primary follicles, while granulosa cells exhibit an upregulation of the cell cycle[23]. Given that our dataset consists of whole follicles encompassing both oocyte and granulosa cells, the results reflect the aggregate changes in oocytes and granulosa cells.

Although the overall gene expression changes were similar in child and adult follicles, we also observed differing patterns (examples of differing patterns presented in Fig. 3D and all patterns in Supplementary Fig. 4A, B). To investigate the possible implications of this, we analyzed the genes belonging to major differing patterns. The major adult differing pattern (pattern 20, $n = 56$ DEGs, Fig. 3D) was found to be involved in ribosomal processes and translation (Supplementary Data 5C). Lastly, the major child differing pattern (pattern 10, $n = 52$ DEGs, Fig. 3D enriched cell communication and developmental pathways (Supplementary Data 5D).

We also observed changes in miRNA expression during follicle growth in adults and children (Fig. 3G, Supplementary Data 3B). Specifically, miRNA expression changed the most between the primordial and secondary follicle stages in adults ($n = 44$ DE miRNA), while most of the changes were observed between the primordial and intermediate and primordial and primary stages in children ($n = 46$ and $n = 41$ DE miRNAs, respectively) (Fig. 3G). The majority of the DE miRNAs characterizing follicle development from primordial to secondary stage in children and adults were significant only in one age group (Fig. 3H). Over-representation analysis was carried out to identify targets of these miRNAs. The 61 adult DE miRNAs were found to target PI3K-Akt, FoxO, and HIF-1 signaling pathways (Fig. 3I, Supplementary Data 5E). The 70 child-specific DE miRNAs were significantly enriched for multiple pathways encompassing cytokines, PI3K, HIF-1, Jak-STAT, and FoxO signaling (Fig. 3I, Supplementary Data 5F). In addition, multiple cancer-related terms were found in the miRNA target analyses, which likely reflects the primary research focus of miRNA studies on cancer. These results suggest that miRNAs may be involved in controlling early follicle growth, with a greater level of predicted miRNA regulatory involvement in child follicles.

In summary, gene expression changes markedly from the primordial to the secondary stage in both age groups and follows a similar overall pattern where the majority of genes are upregulated and related to ECM modifications, and the downregulated genes are associated with the cell cycle. However, distinct subsets of polyA-RNAs and miRNAs exhibited differential expression patterns during follicle development when comparing children with adults, and their impact on follicle biology warrants further investigation.

### Follicles show the highest differences at the secondary stage

We next focused on the differences between child and adult follicles in more detail. PCA of each follicular stage separately revealed the effect of age on polyA-RNA and miRNA profiles (Supplementary Fig. 5A, B). Gene expression variation between adult and child follicles at identical developmental stages was investigated and the largest differences were found in the secondary stage both in polyA-RNA (387 DEGs, Supplementary Data 6A–D) and miRNA (11 DE miRNAs, Supplementary Data 8A–C) expression levels (Fig. 4A). To assess the robustness of our differential expression analysis results, we conducted a bootstrapping analysis. The results displayed that the same DEGs were detected with high frequency across the 200 bootstraps, suggesting that they were not detected by chance (Supplementary Data 6A–D). GO and pathway enrichment analyses were performed on the pooled DEGs for adults

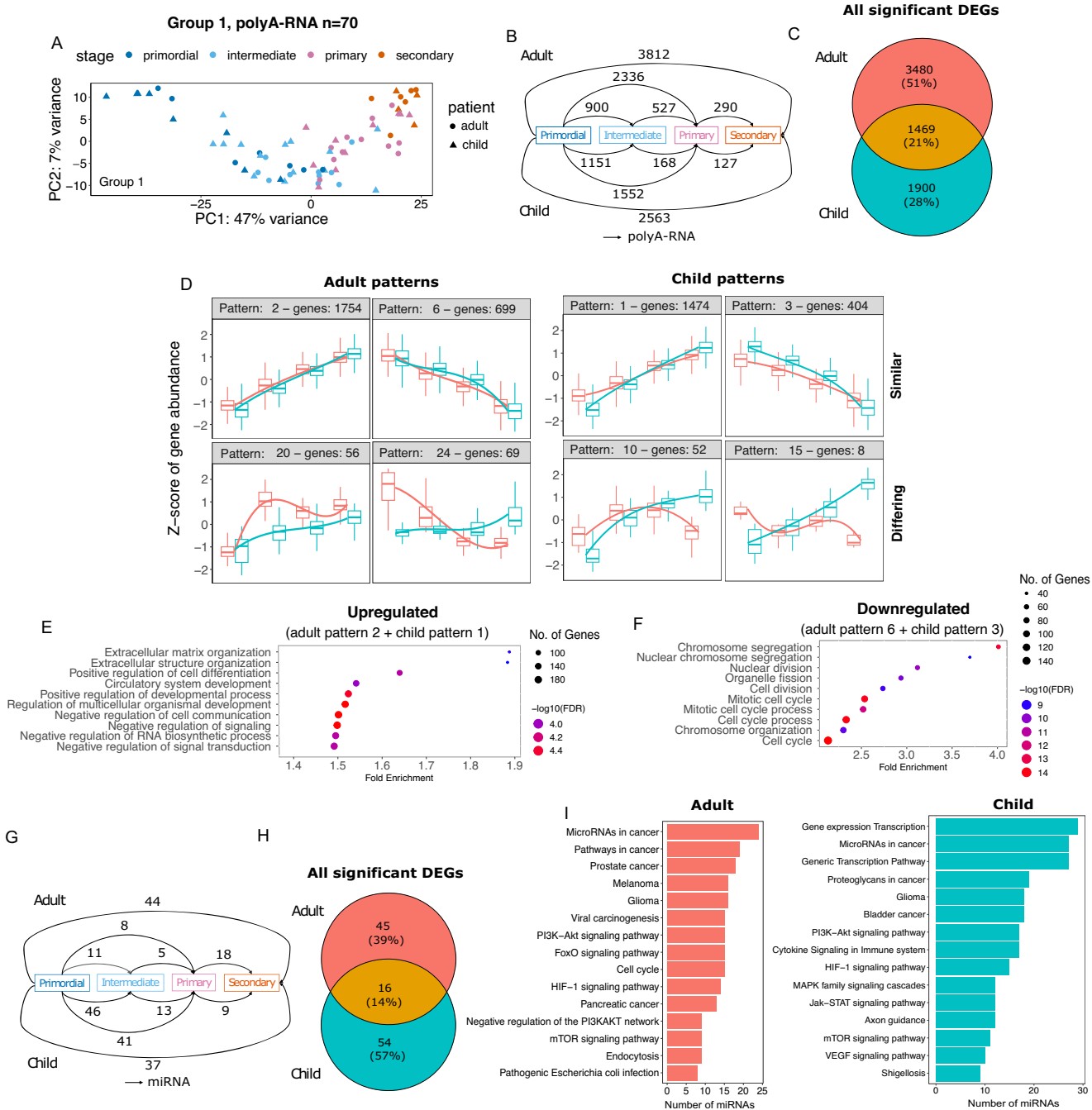

**Fig. 3 | Transcriptional dynamics of primordial to secondary follicle development in children and adults.** Gene expression changes during follicle development up to the secondary stage in children and adults were studied using data from Group 1 follicles (*n* = 70). **A** Principal component analysis showed that PC1 was primarily driven by the morphologically determined maturation stage of the follicles in both age categories. **B** Comparative analysis of different follicular stages revealed thousands of differentially expressed genes (DEGs, FDR < 0.05) in children and adults, with many of these changes occurring during the transition from dormancy (primordial stage) to active growth (primary stage). **C** Comparison of DEGs between children and adults revealed minor overlap, with the majority of DEGs being significant only in one age group. **D** Separate pattern analysis based on DEGs in adults and DEGs in children revealed that most genes were upregulated, while some were downregulated. The corresponding genes in the other age group are shown for comparison in the plots, demonstrating that the overall regulations are similar despite statistical testing suggesting the genes are regulated in one age group only. Despite these overall similarities, multiple smaller groups of genes

showed differing patterns (bottom row). Comprehensive patterns can be found in Supplementary Fig. 4. In box plots, the center line represents the median, the hinges correspond to the first and third quartiles (interquartile range), and the whiskers extend to 1.5 times the interquartile range from the hinges. **E** Genes displaying upregulation during follicle development (adult pattern 2 + child pattern 1) were enriched in Gene Ontologies (GO) related to extracellular matrix organization and developmental processes. **F** Conversely, genes exhibiting downregulation (adult pattern 6 + child pattern 3) were linked to GO categories related to cell cycle processes, chromosome segregation, and cytoskeleton organization. **G** Changes in miRNA levels were comparatively subtle. **H** The majority of miRNAs were significant only in one age group. **I** Predicted adult miRNA targets showed over-representation in the PI3K-Akt, FoxO, and HIF-1 signaling pathways, while child miRNAs displayed over-representation in the PI3K-Akt, HIF-1, Jak-STAT, and VEGF pathways. All DEGs have been adjusted for patient-specific variation. Source data are provided as a Source Data file.

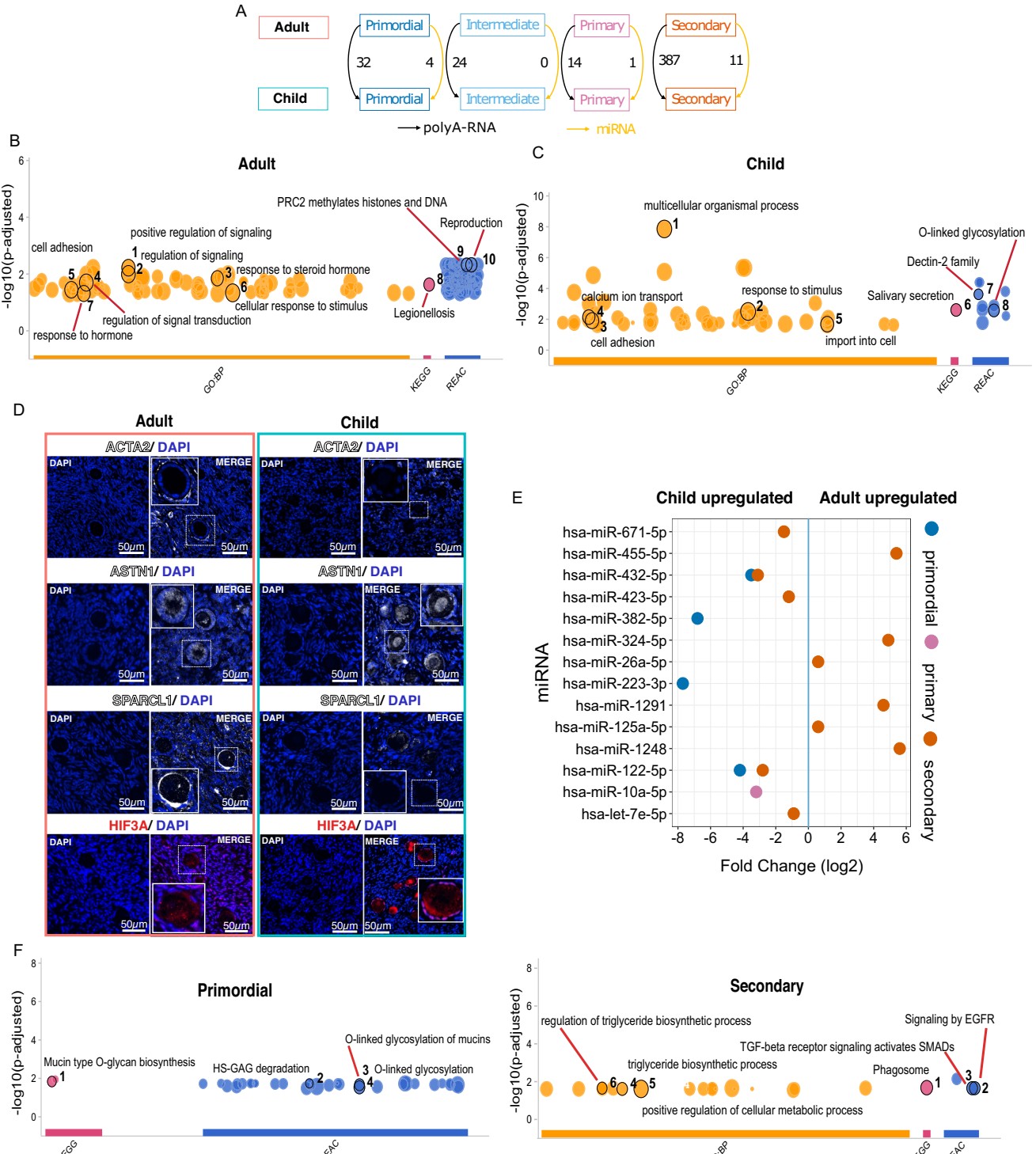

**Fig. 4 | Divergence in gene expression between child and adult follicles.** Gene expression variation between child and adult follicles at corresponding developmental stages was examined. **A** Comparison of the same-stage follicles revealed the most pronounced differences between the age groups at the secondary stage in both polyA-RNA and miRNA expression. **B** Top enriched Gene Ontologies (GO), KEGG, and Reactome pathways in adult follicles pertained to hormones, signaling, cell adhesion, and reproduction, Fisher's one-tailed test, adjusted for multiple comparisons using Benjamini–Hochberg (BH) method, while **C** differentially expressed genes (DEGs) upregulated in child follicles were related to calcium ion transport, adhesion, and glycosylation, Fisher's one-tailed test, adjusted for multiple comparisons using BH method. **D** Immunostaining of selected DEGs belonging to the top enriched processes of signaling (*ACTA2*, increased in adult),

development and adhesion (*ASTN1*, increased in child; *SPARCL1*, increased in adult), and development and morphogenesis (*HIF3a*, increased in child), validated the anticipated differences at the protein level (adult $n = 2$ and child $n = 2$). **E** Dot plot displaying the differentially regulated miRNAs. The profile transitioned from an upregulation of miRNAs in child primordial follicles to an upregulation of an alternate set of miRNAs in adult secondary follicles. **F** In primordial follicles, predicted targets exhibiting negative correlations with miRNA expression values were enriched in O-linked glycosylation of mucins pathway, Fisher's one-tailed test, adjusted for multiple comparisons using BH method. Additionally, in secondary follicles, negative correlations were observed in the context of TGF-beta signaling and the triglyceride biosynthetic process (for a comprehensive list, see Supplementary Data 10A, B). Source data are provided as a Source Data file.

and children separately (FDR < 0.05, fold change >1.5). Interestingly, DEGs upregulated in adult follicles across the developmental stages displayed enrichment in hormones, signaling, cell adhesion, and reproduction (Fig. 4B, Supplementary Data 7A), whereas calcium ion transport, cell adhesion, and O-linked glycosylation pathways were linked to DEGs upregulated in child follicles (Fig. 4C, Supplementary Data 7B).

DEGs from the top significant gene sets were filtered according to their potential roles in ovaries (Supplementary Data 7), and immunostaining was used to validate the presumed differences in expression at the protein level. Interestingly, the most significantly enriched gene sets contained multiple markers that have also been used to identify theca cells. For example, Astrotactin 1 (*ASTN1*) and SPARC Like 1 (*SPARCL1*), which relate to cell adhesion and development, are known to be associated with theca cells in opposite expression directions during follicle development; *ASTN1* is downregulated, and *SPARCL1* upregulated[28,29]. Likewise, Actin Alpha 2, Smooth Muscle (*ACTA2*), which appeared in multiple gene sets related to development and signaling, has also been shown as an early theca cell marker[22]. Finally, multiple gene sets related to development and morphogenesis that were enriched in child follicles included the Hypoxia-Inducible Factor 3 Subunit Alpha (*HIF3A*). HIFs are known to regulate adaptive cellular responses to decreased oxygen levels[30], and may play a role in follicle dormancy in the poorly vascularized cortex[31]. These four markers were selected for immunostaining experiments. An independent set of ovarian tissue sections were stained (child fertility preservation patients $n = 4$, c-sec $n = 1$, and GRP $n = 3$) (Supplementary Data 1C), and the results largely confirmed the expected differences at the protein level. ACTA2 and SPARCL1 displayed higher expression in adult tissue sections, and HIF3α was expressed at a higher-level expression in child follicles (Fig. 4D). ASTN1 levels displayed inter-individual variation between patients and the expected pattern was only found in some samples (Fig. 4D).

miRNA levels differed between child and adult follicles, mainly at the primordial ($n = 4$) and secondary stages ($n = 11$) (Fig. 4A and Supplementary Data 8). Interestingly, all DE miRNAs in primordial and primary follicles were upregulated in child follicles, while the majority of DE miRNAs in secondary follicles were upregulated in adults (Fig. 4E, Supplementary Data 8). Given that the data on polyA-RNA and miRNA were derived from the same follicles, their expression levels can be correlated. We, therefore, investigated correlations between DE miRNA and their predicted mRNA targets in Group 1 follicles, which were differentially expressed between children and adults. In total, we identified 8 DE miRNAs that exhibited a significant negative correlation with 23 predicted target genes (Supplementary Fig. 6, Supplementary Data 9). To assess the potential processes influenced by miRNA regulation, we performed GO and pathway analyses. The miRNA targets in child primordial follicles were involved in various glycosylation reactions, whereas miRNA targets in adult secondary follicles were associated with triglyceride biosynthesis and TGF-beta signaling (Fig. 4F, Supplementary Data 10). TGF-beta signaling has a well-established role in follicle growth in adults[32], and triglycerides may reflect enhanced energy metabolism and steroidogenesis in secondary follicles[33]. Conversely, the relationships between O-linked glycosylation, mucins, and follicle biology in children remain more elusive, and merit further studies.

To summarize, the differences between child and adult follicles appear to relate to ECM where recruitment of early theca cells is absent or delayed in child follicles compared to adults. In addition, child primordial follicles express elevated levels of miRNAs targeting genes involved in glycosylation, while adult follicles turn on the expression of miRNAs regulating lipid metabolism and TGF-beta signaling as they grow. Since human oocytes employ higher-level of piRNAs compared to miRNAs, the miRNA-related regulations most likely take place in granulosa cells[34].

## Molecular signatures of primordial follicles

Follicle staging is based on morphological appearance that is expected to correlate with the developmental stage of the follicle. We noticed heterogeneity in the transcriptomes of same-stage follicles, even when they all belonged to the same group (e.g., Group 1). For example, the PCA in Fig. 3A shows that while most primordial follicles are close together, some tend to cluster with intermediary follicles. Primordial follicles are the dormant, prenatally formed follicles that constitute the ovarian reserve[3]. They are morphologically characterized by an oocyte surrounded by a thin single layer of flat granulosa cells. Interestingly, PCA of primordial follicles showed that PC1, explaining 31% of the variation in the data, divided the follicles into two clusters containing both child and adult follicles (Fig. 5A). Morphologically, follicles in both clusters seemed similar (Fig. 5B).

We compared gene expression profiles between these two clusters of primordial follicles and found 636 DEGs (FDR < 0.05, Fig. 5C and Supplementary Data 11). Most of these DEGs were upregulated in the cluster that was mixed with intermediary follicles (Fig. 5C). The upregulated DEGs included WT1 Transcription Factor (*WT1*), *FOXL2*, Yes1 Associated Transcriptional Regulator (*YAP1*), Fibroblast Growth Factor 2 (*FGF2*), Epidermal Growth Factor Receptor (*EGFR*), and Early Growth Response 1 (*EGR1*) (Fig. 5D), all which are well-known for their associations with follicle activation and/or growth[35–40]. In addition, GO results displayed (DEGs filtered by FDR < 0.05, fold change >1.5) enrichment in the regulation of cell population proliferation and cell motility (Fig. 5E). Furthermore, the immunofluorescence staining of adult ovarian tissue ($n = 1$ c-sec patient) displayed that some primordial follicles demonstrated upregulated YAP1 within the granulosa cell layer, while others did not (Fig. 5F). Together, these results suggest that some of the primordial follicles had already been activated to grow according to their gene expression, therefore, we refer to them as activated primordial follicles. To conclude, the results illustrate that staging follicles correctly as either dormant or active solely based on morphology is challenging and would benefit from a combination of histology and gene expression analysis.

## Chemotherapy has a dose-dependent effect on gene expression

Ovarian tissue is seldom available for research from healthy donors. Our study was based on ovarian tissue from children undergoing OTC due to upcoming high-risk gonadotoxic treatments, and tissues from adults were collected from gender reassignment surgeries and c-sections. Many of the pediatric patients had already received their first-line treatments before OTC, mainly alkylating chemotherapy, which is very typical for this patient group[41]. Similarly, all GRPs had received androgen therapy as standard care prior to oophorectomy. We tested whether these treatments had any detectable impacts on the transcriptomes of the Group 1 follicles. The length of androgen therapy did not have major effects on follicle gene expression. This is in line with our previous study, where single-cell profiles of ovarian cortical tissue derived from c-sec patients and GRPs did not differ markedly[21]. In addition, ~30% of GRPs on testosterone show signs of ovulation, suggesting normal ovarian function despite the hormone therapy[42]. Nevertheless, it is imperative to acknowledge the potential existence of a nuanced androgenic impact on pre-antral follicles. Our study's limited sample size may have hindered our ability to discern such subtleties. In contrast, gonadotoxic treatments influenced gene expression in the follicles. We counted the cumulative dose of alkylating agents as the cyclophosphamide equivalent dose (CED)[43], and PC2 aligned the follicles at varying stages according to their CED level (Fig. 6A). Such a trend was not present in the miRNA data (Supplementary Fig. 7A). CED doses equivalent to or higher than 6000 g/m² are considered as high risk to ovarian reserve and, therefore, constitute as an indication for fertility preservation[41,44]. We therefore tested if gene expression in follicles that were exposed to high-risk treatment differed from those that were exposed to lower levels and

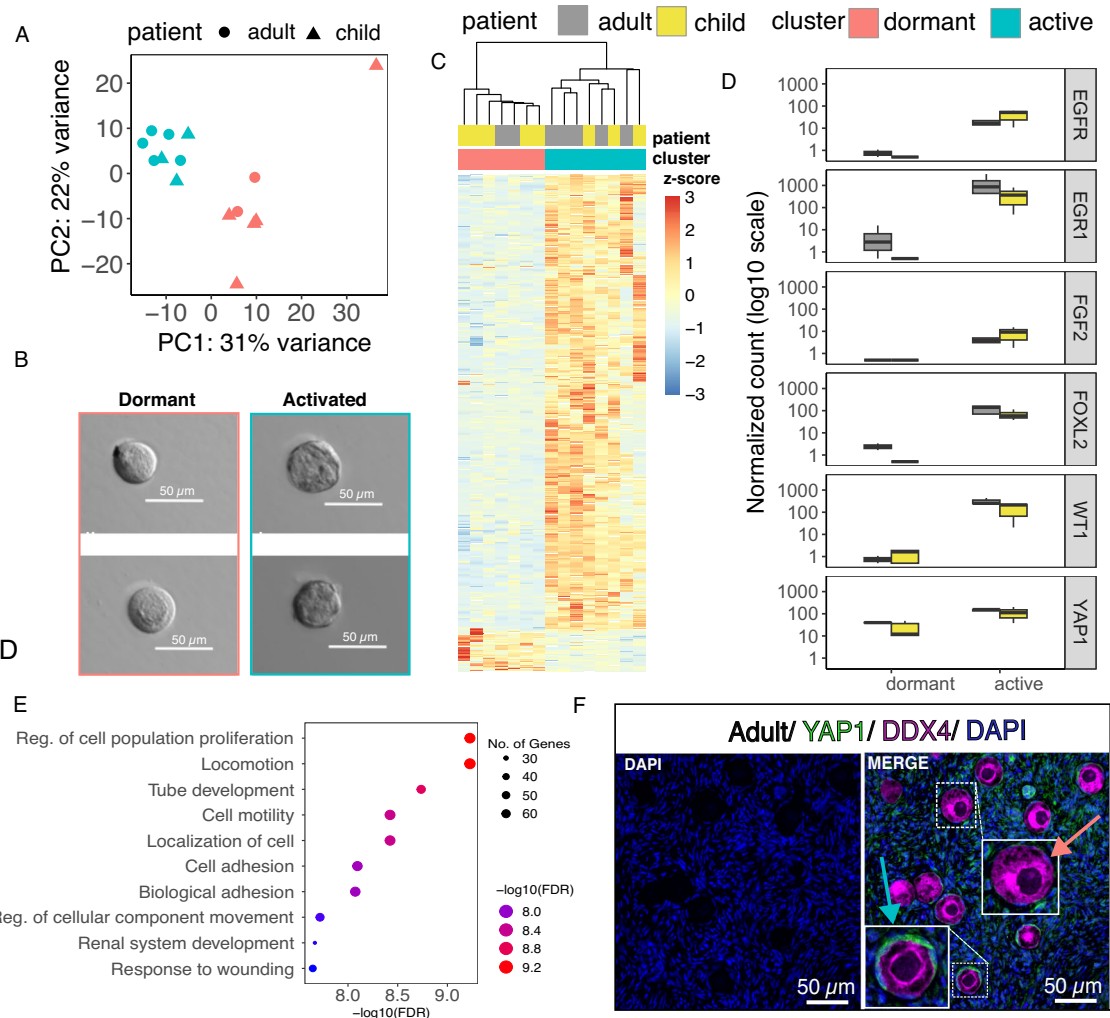

**Fig. 5 | Heterogeneity in gene expression of morphologically similar primordial follicles.** Gene expression variation within the set of 15 primordial follicles was further inspected. **A** Principal component analysis (PCA) based on polyA-RNA expression divided the primordial follicles into two clusters by PC1, with child and adult follicles present in both. **B** In terms of morphology, the follicles from the two clusters were indistinguishable (adult $n = 7$ and child $n = 8$). **C** The two clusters differed by 636 differentially expressed genes (DEGs), with the majority being upregulated in one of the clusters. **D** The top DEGs included many genes with known roles in follicle growth and activation (adult $n = 7$ and child $n = 8$). In box plots, the center line represents the median, the hinges correspond to the first and third quartiles (interquartile range), and the whiskers extend to 1.5 times the interquartile range from the hinges. **E** The associated enriched gene ontologies related to the regulation of proliferation, development, motility, and adhesion, suggesting that some of the primordial follicles had exited dormancy despite not showing morphological signs of activation. **F** Immunofluorescence staining of adult ovarian tissue ($n = 1$) with antibodies targeting YAP1 suggested the presence of dormant (no staining, red arrow) and activated primordial follicles (positive staining in granulosa cells, blue arrow). Source data are provided as a Source Data file.

found 166 DEGs (FDR < 0.05). The DEGs (FDR < 0.05, fold change >1.5) were enriched in interferon signaling pathway genes (Fig. 6B), multiple of which showed significant positive correlation with the CED level: Interferon Induced Protein 44 Like (*IFI44L*), Interferon Alpha Inducible Protein 6 (*IFI6*), Interferon Induced Protein With Tetratricopeptide Repeats 1 (*IFIT1*), Interferon Regulatory Factor 9 (*IRF9*), ISG15 Ubiquitin Like Modifier (*ISG15*), 2′–5′-Oligoadenylate Synthetase 2 (*OAS2*), and XIAP Associated Factor 1 (*XAF1*) (Fig. 6C). Immunofluorescent staining in an independent set of samples ($n = 2$ child fertility preservation patients) confirmed upregulation of IFI44L in treated patient follicles compared to untreated patients (Fig. 6D).

We did not observe differences in classical DNA damage genes (*RAD9, PARP1, BRCA1, ATM,* and *TP53*[45]) between untreated and treated follicles (Supplementary Fig. 7B). Previous experiments with adult and fetal ovarian tissue xenograft models have shown that exposure to alkylating chemotherap -induced DNA damage response in ovarian follicles[46] and a reduction in primordial follicle densities as early as

12–24 hours after exposure[47]. The lack of DNA damage response markers in our dataset could depend on the time since exposure. In our study, ovarian tissue was collected 1-44 days after the last chemotherapy dose, while the time since the first chemotherapy exposure varied from 26 to 113 weeks. These time frames are long, and the initial DNA damage response could have already disappeared. Instead, our data suggest upregulated interferon signaling as a longer-term side effect of chemotherapy in follicles.

We further investigated whether Group 2 follicles were affected by chemotherapy in a manner similar to Group 1 and found comparable impacts on interferon signaling (Supplementary Fig. 7C). Subsequently, we assessed the impact of CED exposure on the predicted signaling functions of Group 2 follicles. Due to the limited sample size, we combined all follicle stages before categorizing them into Group 2 high (6000 g/m² or above) and Group 2 low (<6000 g/m²) exposure groups. Intriguingly, higher CED exposure did not appear to reduce signaling activity in Group 2 follicles, suggesting that their

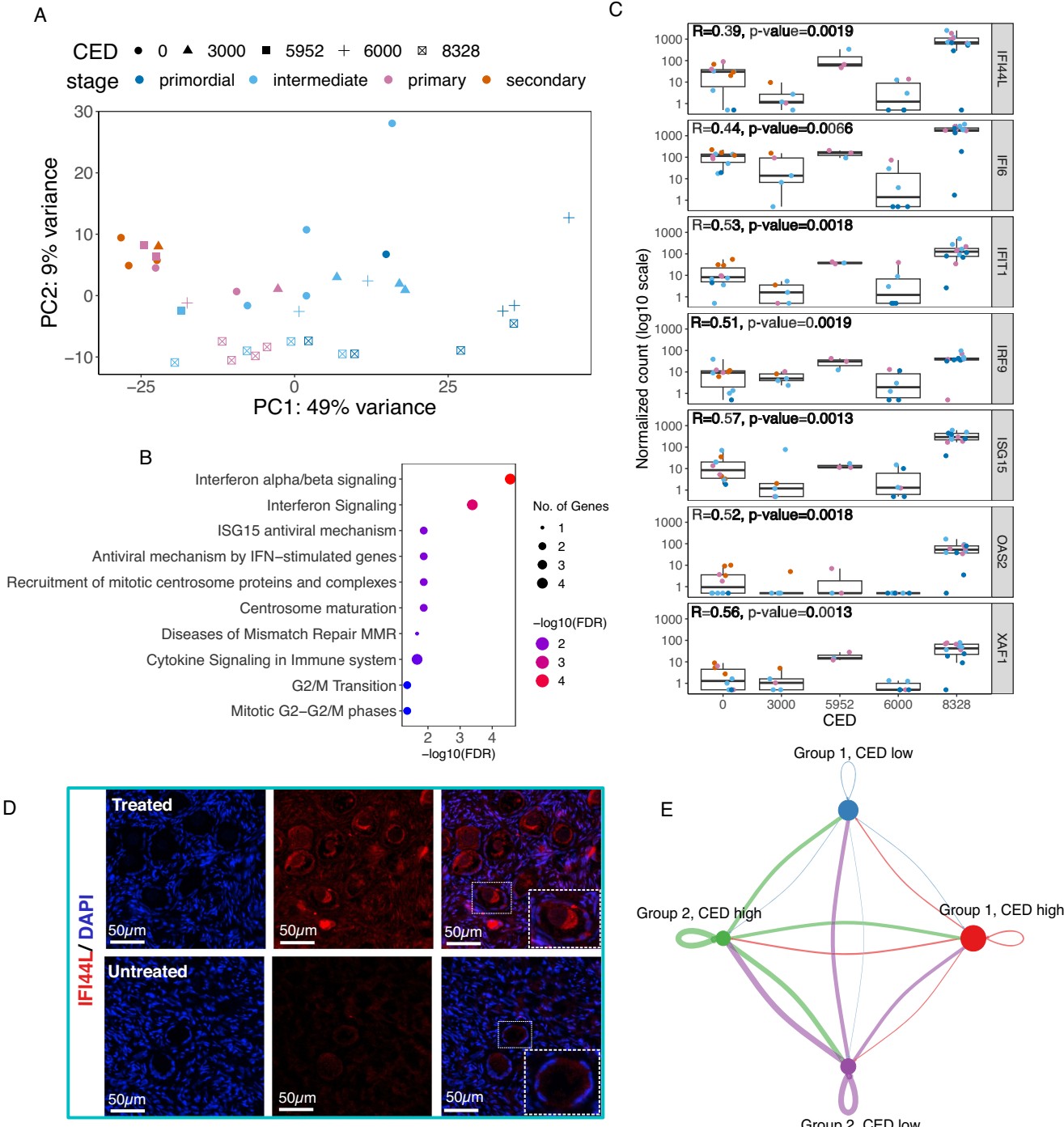

**Fig. 6 | Chemotherapy alters gene expression in viable child ovarian follicles of good morphology.** Since some child patients had received first-line chemotherapy prior to fertility preservation, the potential effects of the cumulative dose of alkylating chemotherapy (CED, cyclophosphamide equivalent dose) on follicle gene expression were examined. **A** Principal component analysis displayed the expected division of follicles by developmental stage across PC1, while PC2 distributed the follicles by the level of chemotherapy treatment. **B** When comparing gene expression between patients at low risk of gonadal damage (CED 0–3000 g/m²) to those at high risk (CED ≥6000 g/m²), 166 differentially expressed genes were identified with significant enrichment of pathways related to interferon signaling. **C** Expression of selected interferon signaling pathway genes correlated with CED exposure levels based on Spearman coefficients (CED 0 $n$ = 10; CED 3000 $n$ = 5; CED 5952 $n$ = 3;

CED 6000 $n$ = 6; and CED 8328 $n$ = 12). In box plots, the center line represents the median, the hinges correspond to the first and third quartiles (interquartile range), and the whiskers extend to 1.5 times the interquartile range from the hinges. **D** Immunofluorescence staining for IFI44L in an independent set of samples (untreated, 6 years old, CED = 0; treated, 6 years old, CED = 4200 g/m²) confirmed upregulation in follicles at the protein level. **E** The circle plot illustrating follicle-to-follicle communication based on secreted signals reveals that the communication strength of Group 2 follicles is comparable under both low and high CED exposure. The line thickness indicates the strength of the communication, color specifies the group, and the size of the dots relates to the number of follicles in each group. Box plots display the median, interquartile range, and minimum and maximum (whiskers) values of the normalized count. Source data are provided as a Source Data file.

hypothesized role in paracrine regulation might remain unaffected at these exposure levels (Fig. 6E).

## Discussion

The unanticipated clear division of the analyzed follicles into two main groups was notable given that, to date, follicles have been classified based solely on their morphology. Our data suggest that morphology alone is not sufficient for follicle classification into further subtypes. The first subtype identified by our study (Group 1) included cortical follicles at all developmental stages displaying the expected oocyte and granulosa cell markers, and the second group (Group 2) consisted exclusively of growing follicles and lacked classical oocyte markers. This might imply that Group 2 follicles are unable to complete folliculogenesis, possess a lower quality in comparison to Group 1, and are approaching degeneration. However, based on the viability staining, morphology, and gene expression, they appeared healthy. Furthermore, Group 2 follicles constituted a third of all cortical follicles; if genuinely atretic, this degree of degeneration would rapidly deplete the ovarian reserve. However, if Group 2 follicles cannot mature their oocyte, they will eventually die. It is plausible that these follicles, not selected for oocyte maturation, remain active via their granulosa cells for an extended period, perhaps to fulfill local functions within the ovary. Corroborating this theory, our data suggest that the granulosa cells of Group 2 follicles are healthy because they did not differ from Group 1 in their miRNA signatures, which is likely derived from the granulosa cells[34]. Furthermore, Group 2 follicles demonstrated significant potential to communicate via many well-known ovarian growth factors, such as AMH and KIT[48,49], in addition to not yet characterized signaling substances like MK and SEMA-3. In summary, we hypothesize that Group 2 follicles play a role in supporting the growth of other follicles before they themselves eventually degenerate. Further studies should investigate whether factors related to oocyte quality, such as chromosomal integrity, influence follicle development towards Group 1 or Group 2.

It is well-documented that follicles grow better when cultured in groups. For instance, the growth of mouse primary follicles in vitro correlates with the number of follicles co-cultured in the same well[50]. Similarly, co-cultured human secondary follicles gave rise to mature oocytes while oocytes from individual secondary follicle cultures failed to mature[51]. Furthermore, primordial follicles in the human ovarian cortex typically co-localize into clusters[52]. Notably, variations in growth patterns between morphologically similar follicles in vitro have been reported in mice and humans, as some follicles grow rapidly, others slowly, and some not at all[53,54]. Consequently, there may be follicles with distinct roles within the ovarian cortex, and successful follicle growth may rely on support from these neighboring follicles. After all, only a few hundred out of a million primordial follicles will ever progress to maturity.

A previous transcriptomic study on individual human pre-antral follicles did not identify distinct follicle groups analogous to Group 1 and Group 2. Specifically, Zhang et al.[23] analyzed the transcriptome of oocytes and granulosa cells separately, at a single-cell level, covering multiple pre-antral stages. Given that oocytes and granulosa cells were sequenced independently, it is conceivable that oocytes with low oocyte marker expression might have been excluded during the data quality control process, thus excluding the possibility of oocyte type divergence.

Interestingly, in mice, two follicle types have been documented[55]. The granulosa cells of mouse primordial follicles stem from two separate origins during early ovary development (epithelial and mesenchymal origins), leading to the formation of first and second-wave follicles; the former relating to puberty onset, and the latter to the sustenance of long-term fertility[55]. It is possible that human Group 1 and Group 2 follicles are analogous to mouse first and second-wave follicles. However, the relative proportions of Group 1 and Group 2

follicles remained consistent from childhood to adulthood, which is contrary to the expected trend based on the mouse studies. Moreover, although detailed analyses are missing, it seems that both the first and second-wave follicles express oocyte markers in mice[55]. Future studies should directly compare isolated human and mouse follicles at comparable stages.

Transcriptomic changes from the primordial to the secondary follicle stage were marked in both children and adults, entailing thousands of genes. The largest change occurred during the primordial-to-primary transition, as previously reported in mice[56]. The overall changes related to ECM remodeling, which is a process essentially linked with folliculogenesis[17,19]. Unlocking the mechanisms controlling follicle dormancy and activation could facilitate innovations aimed at regulating fertility. While thousands of genes are difficult to target, we found a smaller group of genes that may reflect the first stage of activation in primordial follicles, where morphological signs of activation were still absent. These early activation markers included effector protein of the Hippo signaling pathway (YAP1) and known follicular transcription factors (FOXL2, WT1). Modulation of these genes could prolong the fertile lifespan by slowing down follicle activation, or boost the growth of follicles, for example, in advanced-aged women wishing for pregnancy.

The gonadotropin-independent follicles in the cortex could be assumed to be the same before and after puberty, as these fetally formed primordial follicles remain dormant for decades and the early steps of growth are not dependent on pituitary hormones. Yet, for example in mice, the oocytes obtain their appropriate chromatin configuration under the influence of FSH after puberty[57]. In addition, ovarian ECM composition differs between adults and children[17], which could also relate to FSH, as this hormone can stimulate granulosa cells to secrete proteases, their inhibitors, and angiogenic growth factors[58,59]. A combination of hormonal and ECM differences could lead to differential gene expression between child and adult follicles. Indeed, we found that child and adult follicles significantly differed in gene expression. Child-specific miRNA and their predicted targets were enriched for glycosylation reactions, mucins, interleukins, and signaling, while adult-specific differences related to cell cycle, steroids, fatty acid metabolism, and signaling.

The top enriched processes characterizing differences between child and adult follicles contained ACTA2 and SPARCL1, genes known as early theca cell markers[22,29]. Theca cell recruitment starts soon after follicle activation, and our data suggest that child follicles may not initiate this process in a timely manner. Interestingly, granulosa-derived growth factors known to be involved in theca cell recruitment (KITLG, FGF2, LIF, and FGF7)[60] showed no differences between children and adults in our dataset, suggesting that observed differences may be attributed to the surrounding stroma rather than intrinsic follicular signals. For example, the higher expression of mucins in child follicles could modify cell adhesion and growth factor signaling, potentially leading to altered theca cell recruitment.

In addition to theca cell-related genes, the hypoxia family gene HIF3A displayed higher expression in child follicles at both RNA and protein levels. HIF3A is one of the hypoxia-inducible factors (HIFs), which regulate adaptive cellular responses to decreased oxygen levels. In fertility preservation, a significant loss of follicles typically occurs shortly after the transplantation of tissue, presumably due to ischemic damage[61]. The differing expression of HIF3a between child and adult follicles could suggest differing sensitivities to changes in oxygen supply, which may be clinically relevant when ovarian tissue from children is transplanted. The role of hypoxia in follicle dormancy, activation, and death, especially before and after puberty, should be further elucidated.

Fertility preservation patients are heterogenous in terms of age, diagnosis, and treatments. Initial treatment plans may be of low risk; however, upon cancer progression or relapse, the cumulative CED can

reach levels associated with a high risk for infertility[62]. In our study, the child follicles were derived from patients undergoing fertility preservation, with the highest doses (CED ≥ 6000 mg/m²) carrying a high risk of infertility as a late side effect[62]. Although it is established that high cumulative CED leads to follicular death, the specific targets of alkylating chemotherapy in human ovaries have remained elusive[63]. We found elevated expression of interferon pathway genes as a function of CED. Interferons are known for antiviral and immunomodulatory effects[64], and agents like cyclophosphamide can activate interferon signaling and induce a sterile inflammatory response[65]. This immunostimulatory effect may be advantageous for antitumor response and clearance[66], but interferon signaling can also lead to atresia in follicles[67]. The follicles in our study were still viable at collection. With the time since treatment varying from days to months in our study, it is uncertain whether the analyzed follicles with elevated interferon signaling would have eventually died. Inhibiting interferon signaling to protect the ovarian reserve during chemotherapy is problematic as the interferon response plays a role in tumor clearance[66]. Alternatively, protecting follicles from downstream processes like apoptosis, for example, through inhibition of checkpoint kinases, might mitigate the off-target ovarian damage of chemotherapy[46,68]. Nonetheless, such strategies require rigorous testing to ensure they do not compromise oocyte quality or increase the risk of birth defects.

Our findings repeatedly revealed discrepancies between follicle morphology and its transcriptomic signature. Given that follicle staging based on morphology has long been the gold standard, this observation could bear significant consequences, subject to validation by independent studies. For example, existing follicle growth models rely on mathematical presumptions that all follicles exhibit uniform behavior. The potential presence of two distinct follicle types in the cortex may disrupt these assumptions, subsequently affecting the derived models of the reserve. Secondly, the ovarian reserve is typically determined either by counting follicles in the cortex (when tissue samples are available) or by measuring AMH in blood samples. Whether these metrics accurately reflect the true number of follicles that can mature until ovulation, specifically, the Group 1 follicles as identified in our study, is not clear. Further research is needed to understand how the proportions of Group 1 and Group 2 follicles might change from childhood throughout the reproductive lifespan, whether they vary between individuals or by exposure to external factors, and whether both types would be needed in artificial ovarian constructs.

While our research encompasses data from over 100 meticulously staged follicles from nine patients, and independent validations in eighteen additional patients, it is not without limitations. One significant constraint is the relatively small number of patients and follicles per follicle stage. Obtaining human ovarian tissue, especially from children, presents significant challenges, which limited the number of participants in our study. Additionally, a majority of the child follicle donors had undergone first-line chemotherapy. As a result, we could not fully account for potential variability between patients or the effects of different diagnoses. Nevertheless, we analyzed the data using multiple approaches, ranging from down-sampling to bootstrapping and different normalization methods, always arriving at similar conclusions, suggesting that the findings are robust.

In conclusion, our study is the first one to report polyA-RNA and miRNA transcriptomic profiles in individual prepubertal and adult ovarian cortical follicles. The discernible differences between child and adult follicles pertained to the ECM, theca cells, and hypoxia. Our observations have the potential to refine fertility preservation for children by emphasizing the likely importance of an adult-like growth environment, for example, upon transplantation or in vitro follicle growth. Future endeavors to identify markers for a more refined staging and categorization of human ovarian follicles, as well as understanding their quality and roles in maintaining fertility, will be invaluable. Such efforts will significantly broaden the available options for preserving fertility in girls and women, be it through refinement of protocols for ovarian tissue handling during OTC and transplantation, devising protective therapies during chemotherapy, or advancing the development of artificial ovaries.

## Methods

### Ovarian tissue handling

Adult ovarian tissue collection and use for research were approved by the Stockholm Region Ethical Review Authority (license number 2015/798-31 with amendments) and by the Ethics Committee of Helsinki University Hospital (HUS/3319/2017 and HUS/ 2087/2023). Hospital personnel informed GRPs and c-sec patients about the study and the patients who signed the informed consent were included. Childhood cancer patients or patients with severe hematological disease who were at high or very high risk of iatrogenic premature ovarian insufficiency due to planned treatments (such as allogenic/autologous hematopoietic stem cell transplantation, radiotherapy with ovary in the radiation field, or excessive chemotherapy) were offered OTC in compliance with the guidelines on fertility preservation of the Nordic Society of Pediatric Hematology and Oncology (NOPHO) and Swedish national recommendations for infertility risk assessment (https://vavnad.se/atgarder-for-att-bevara-av-reproduktionsformaga-hos-unga/). The provision of OTC was part of a research protocol in children, which was approved by the Swedish Ethical Review Authority (Sveafertil patients, Dnr:2019-03802) and by the Ethics Committee of Helsinki University Hospital (HCH, Dnr:340/13/03/03/2015). Written informed consent was obtained from all age-appropriate patients or guardians, and parental written consent was obtained for all patients under 18 years old. To ensure privacy, all samples were pseudonymized with random codes in referring hospitals, before the samples became available for research. The study was conducted in accordance with the Declaration of Helsinki. The child patients used in transcriptomic analyses were 1–11 years old. The patients used in validation studies were 2–6 years old. The adult patients used in transcriptomic analyses were 19–29 years old, and the ones used in validation were 22–36 (Table 1, Supplementary Data 1). Participants were not compensated for ovarian biopsy collection.

In the case of the GRP surgery, approximately half of the ovary was obtained. Ovaries were transported to the research laboratory in Dulbecco's phosphate-buffered saline (DPBS, Thermo Scientific). The ovarian cortex was separated from the medulla using scalpels and trimmed until ~1 mm cortical thickness was reached. In the case of child ovarian tissue and samples from c-sec patients, the research was conducted on small biopsies that did not require trimming of medulla. GRP and child ovarian cortex tissue were cryopreserved using a slow-freezing method according to the published protocol[69]. Briefly, cortical tissue was cut into pieces of 5 × 5 × 1 mm in size and equilibrated for 30 min on ice in a slow-freezing medium of 7.5% Ethylene Glycol (Sigma-Aldrich), 33.9 mg/ml sucrose (Sigma-Aldrich) and 10 mg/ml human serum albumin (HSA, Vitrolife) in DBPS (Thermo Scientific) while shaking. After incubation, cortex pieces were transferred to cryotubes containing 1 ml slow-freezing media, and a program was started using a controlled-rate freezer (Kryo 230-1.7, Planer PLC).

Pieces of the ovarian cortex for immunofluorescence evaluation were fixed freshly using 4% methanol-free formaldehyde (Thermo Scientific). Dehydrated cortex pieces were embedded in paraffin.

### Isolation of ovarian follicles

Cryopreserved cortex tissue was thawed by incubating tubes in a water bath at 37 °C for around 1 min. Tissue pieces were transferred into three consequential thawing media (TM) solutions supplemented with 10 mg/ml HSA (Vitrolife) and decreasing sucrose and ethylene glycol (both Sigma-Aldrich) concentrations in DPBS (Thermo Scientific) (1.26% ethylene glycol in TM I, 85 mg/ml sucrose in TM I and II and only HSA in TM III) for 10 min per thawing solution. Thawed cortex tissue

pieces (two 2 × 5 mm pieces per patient) were mechanically and enzymatically dissociated. In brief, McIlwain Tissue Chopper was used until tissue was finely fragmented or for a maximum of 10 min in digestion media. Digestion media compositions differed slightly in the 2 laboratories, where the laboratory in Helsinki used lower concentrations of specific enzymes as mentioned below. Digestion media consisted of McCoy's (Thermo Scientific) media with 1 mg/ml HSA (Vitrolife), 1× GlutaMAX (Thermo Scientific), 1× Insulin-Transferrin solution (Thermo Scientific), 40 µg/ml Liberase (Sigma-Aldrich), 0.2 or 0.4 mg/ml Collagenase IV (Thermo Scientific), and 18 or 80 K units/ml DNase I (QIAGEN) Tissue suspension was then incubated in digestion media containing Neutral Red (59 µg/ml, Sigma-Aldrich) at 37 °C under gentle orbital agitation for 30–50 min, until follicles were visibly separated from the stroma. Digestion was stopped using termination media containing McCoy's with 10% HSA (Thermo Scientific and Vitrolife). Follicles were evaluated under a stereomicroscope (Leica S9D or M165C) and only viable follicles were picked. Individual isolated follicles were photographed under inverted microscopes (Olympus CKX41SF or Nikon Eclipse TE2000-S) and classified by their maturation stage directly by two independent observers. All isolated follicles underwent a second round of classification, based on the photographs, by the same observers. Depending on the oocyte maturational stage, oocyte nuclei could be detected, and granulosa cells could be identified either on top or on the side of the oocyte by the difference in depth and cell size. Primordial follicles were staged when flat granulosa cells were identified (Supplementary Fig. 2A). Intermediary follicles were identified as an oocytes with a mixture of flat and cuboidal cells. Primary follicles were identified when no flat granulosa cells could be seen and ~1 layer of cuboidal granulosa cells surrounded the oocyte. Secondary follicles were identified when more than 1 layer of granulosa cells surrounded the oocyte (Supplementary Fig. 2). Isolated follicles were then transferred to tubes containing 4 µl of lysis buffer containing 0.2% Tween 20 (Merck), 2 U/µl RNase inhibitor (Invitrogen) in RNase-free water (Invitrogen) and stored at −80 °C until library preparation.

## Smart-seq2 library preparation and sequencing

Full-length RNA libraries were prepared according to the Smart-seq2 protocol[70] with the following modifications. The initial RNA for the library preparation was 2 µl of follicle cell lysate (adult $n = 60$ and child $n = 60$). Before removing 2 µl of cell lysate from the tubes, the lysate was heated at 72 °C for 3 min. The cell lysate was placed in a 96-well plate (Eppendorf) and 2 µl lysis buffer together with 1 µl 10 mM dNTP mix and 1 µl freshly diluted 10 µM Oligo-dT30VN (Metabion) were added. In reverse transcription reaction, RNase-free (Invitrogen) water volume was 0.59 µl to reach a reaction volume of 6 µl, which then was added to the 4 µl cell lysate to obtain 10 µl reaction volume. cDNA preamplification included 5.5 µl 5× Phusion GC buffer and 0.55 µl Phusion enzyme (2 U/µl) (Thermo Fisher Scientific), 0.825 µl DMSO (AppliChem), 0.55 µl 10 mM dNTP mix and 0.275 µl freshly diluted 10 µM ISPCR (Metabion) primer in 8.8 µl RNase-free water (Invitrogen). PCR products were purified using AMPure XP beads (Beckman Coulter) in 1:0.9 ratio. DNA concentrations and DNA integrity were measured with Qubit HS DNA assay (Invitrogen) and Agilent HS DNA assay (Agilent Technologies), respectively. The tagmentation reaction was carried out by using the Illumina Nextera XT DNA preparation kit (Illumina). Tagmentation reaction included 2 µl ATM and 4 µl TD buffer. To the tagmentation reaction, 1 ng of cDNA was added with final volume of 10 µl. Amplification of adapter-ligated fragments was carried out using 6 µl Nextera PCR Master Mix, 2 µl i5 primer, and 2 µl i7 primer (both from TAG Copenhagen) using 12 cycles to amplify the DNA. The amplification reaction was purified using AMPure XP beads in a 1:0.9 ratio. Final DNA library was quality controlled using Qubit HS DNA assay (Invitrogen) and Agilent HS DNA kit (Agilent Technologies). Libraries were sequenced in equimolar amounts using single-end

76 bp with dual index (8 bp) sequencing on NextSeq 500 platform with NextSeq HIGH75 kit (Illumina).

A separate set of follicles (adult $n = 39$ follicles and child $n = 8$ follicles) were sequenced as described above apart from 4 µl of starting cell lysate volume. The distribution of the follicle stages is presented in Supplementary Data 1B.

## Small RNA library preparation and sequencing

Small RNA libraries were prepared with QIAseq miRNA Library Kit (QIAGEN) according to the manufacturer's protocol. The starting amount of small RNA in library preparation was 2 µl of follicle cell lysate (adult $n = 60$ and child $n = 60$). Final libraries were separated and excised from 5% TBE agarose gels (Bio-Rad) after staining with 1× SYBR Gold stain (Thermo Scientific). Gel pieces containing the small RNA libraries were crushed with pellet pestles (Fisher Scientific). 300 µl RNase-free water (Invitrogen) was added to the gel debris and rotated for 2 h at room temperature to elute small RNA libraries. Eluate and gel debris were transferred to the Spin X centrifuge tube filter (Merck) and centrifuged for 2 min at 16,000 × g. Thereafter, 2 µl glycogen (Thermo Fisher Scientific), 30 µl 3 M NaOAc (Thermo Fisher Scientific), 1 µl 0.1× Pellet Paint (Merck) and 975 µl of cold 100% ethanol (Naxo) were added to the eluate and centrifuged for 20 min at 20,000 × g at 4 °C. Pellet was washed with 500 µl 70% ethanol and centrifuged for 2 min at 20,000 × g. The final libraries were resuspended in 7 µl resuspension buffer (PerkinElmer). The size of libraries was estimated with Agilent DNA High Sensitivity chips on the Agilent 2100 Bioanalyzer system (Agilent Technologies). Library concentrations were measured using Qubit High Sensitivity Assay kit (Thermo Scientific) before pooling in equimolar amounts. Single-end sequencing of 75 bp with single index (6 bp) length was performed on NextSeq 500 platform with NextSeq HIGH75 kit or NextSeq MID150 kit (Illumina).

## Smart-seq2 sequencing data analysis

Raw FASTQ files were quality filtered with Trimmomatic v.0.39 with the option SLIDINGWINDOW:2:20[71]. Reads below 36 nucleotides in length were discarded and all adapters' sequences provided by Illumina were removed from the sequences. The remaining reads were aligned to the primary assembly of human genome GRCh38 retrieved from GENCODE using STAR aligner v.2.7.1 with standard settings[72]. Annotation of reads to genes was performed with htseq-count algorithm[73] (HTSeq v.0.11.2, Python v.3.6.4) by using the primary assembly annotation.gtf file v.35 from GENCODE. The intersection-nonempty option was used in htseq-count for annotating reads that overlapped with multiple features.

The count files produced by htseq-counts were merged using edgeR package v3.36.0[74], and the formed count matrix was used as input for DeSeq2 v.1.34.0[75] in R v.4.1.1 for differential gene expression analysis between groups with standard options. Genes expressed at low levels (below 10 raw counts across all samples) were removed from the analysis. For visualization purposes, variance stabilizing transformation of data was performed with option blind = FALSE. According to the PCA results, 9 outlier samples with low read count and separate clustering from other samples were removed. Additionally, two follicles were removed due to the low-quality microscopy image inhibiting the determination of the correct follicular stage. Therefore, 109 follicles remained for downstream analysis after outlier removal. Stage-wise gene expression differences were calculated using two-factor variables: stage and patient. The patient variable was used to account for potential patient-specific effects on gene expression. Down-sampling was performed using a random selection of 6 or 4 follicles from each stage in the adult and child dataset, respectively. Down-sampling was performed 5 times.

In a downstream analysis of Smart-seq2 data following R packages were used: umap v.0.2.9.0, DEGreport v.1.30.3, ggplot2 v.3.4.0, pheatmap v.1.0.12, EnhancedVolcano v.1.12.0 and CellChat v.1.6.1.

Enriched pathways were detected using a publicly available web application g:Profiler[76] (https://biit.cs.ut.ee/gprofiler/gost) with the following settings: Only annotated genes, Benjamini-Hochberg (BH) FDR < 0.05, GO Biological process, Reactome, and KEGG and visualized using publicly available web application ShinyGO[77] (http://bioinformatics.sdstate.edu/go/).

Follicle-to-follicle communication was based on the CellChat tutorial (https://github.com/sqjin/CellChat). In brief, the loaded count matrix was normalized using normalizeData function (scale factor 10,000) and log-transformed with a pseudo count of 1. A CellChat object was created, and samples were grouped according to the follicle stage, along with grouping information. Group 1 follicles were labeled as primordial1, intermediate1, primary1, and secondary1, while Group 2 follicles were labeled as intermediate2, primary2, and secondary2. The CellChatDB.human database was utilized for ligand-receptor interaction information, and only a subset of Secreted Signaling was employed in the analysis. Communication probabilities were calculated using default settings. Additionally, the aggregated cell-cell communication network was computed by summarizing the communication probabilities. Furthermore, the dominant senders (sources) and receivers (targets) of cell-cell communication were visualized in 2D space. With this information, chord diagrams depicting dominant sources (Group 2 follicles) and dominant targets (Group 1 follicles) were created. In addition, two other count normalization methods (Deseq2 and EdgeR) were tested to confirm that follicle-to-follicle communication results are not dependent on normalization methods.

The Spearman correlation between CED exposure and RNA expression was calculated using normalized counts obtained from DESeq2. To ensure data suitability for analysis, we log-transformed the normalized counts and added a pseudo count of +1. Subsequently, we applied BH correction to adjust the p-values.

### Small RNA-sequencing data analysis

Raw FASTQ files were quality filtered with Trimmomatic v.0.39[71] with the options of SLIDINGWINDOW:2:20. Adapter sequences (3'adapter AACTGTAGGCACCATCAAT and 5' adapter GTTCAGAGTTCTA-CAGTCCGACGATC) were removed and reads below 17 nucleotides in length were discarded. The remaining filtered and trimmed reads were counted and mapped to the primary assembly of human genome GRCh38 using miRDeep2 with standard settings[78] to obtain miRNA sequences from small RNA data.

Count tables from individual samples were merged using edgeR package v.3.36.0[74] and the formed count matrix was used as input for DESeq2 v.1.34.0[75] in R v.4.1.1 for differential gene expression analysis between groups with standard options. miRNAs expressed at low levels were removed from the analysis (below 5 raw counts across all samples). For visualization purposes, variance stabilizing transformation of data was performed with option blind = FALSE. According to the principal component results, 5 outlier samples with low miRNA counts were also removed from data analysis. Additionally, two follicles were removed due to the low-quality microscopy image, inhibiting the determination of the correct follicular stage. Therefore, after outlier removal 113 follicles remained for downstream analysis. Stage-wise gene expression differences were calculated using two-factor variables stage and patient. The patient variable was used to account for potential patient-specific effects on gene expression.

In a downstream analysis of miRNA data, the following R packages were used: ggplot2 v.3.4.0, pheatmap v.1.0.12, EnhancedVolcano v.1.12.0, and Hmisc v.4.7-2.

For Spearman correlation between mRNA and miRNA molecules within the same follicle normalized counts from DESeq2 were used. To ensure data suitability for analysis, we log-transformed the normalized counts and added a pseudo count of +1. Subsequently, we applied BH correction to adjust the p-values.

A list of miRNAs DE expressed between different follicle stages in adult or child samples was used as input for miRNA Enrichment Analysis and Annotation Tool (miEAA[79], https://ccb-compute2.cs.uni-saarland.de/mieaa/), which performs over-representation analysis using linked external databases. The following categories were selected: KEGG (miRPathDB) and Reactome (miRPathDB). The significance level was set at 0.05 and BH adjustment was applied.

### Immunofluorescence staining

Embedded formalin-fixed tissues were sectioned at 4 μm and placed on SuperFrost Plus white slides (VWR) by the Morphological Phenotype Analysis Core Facility (FENO, Karolinska Institute, Sweden). Immunofluorescence staining was then performed on GRP, c-sec, and childhood fertility preservation patients (Supplementary Data 1C). Briefly, sections were deparaffinized and rehydrated prior to staining. Antigen retrieval was then performed using a buffer containing either 10 mM Tris/1 mM EDTA (pH 9, Sigma-Aldrich) or 10 mM sodium citrate tribasic dihydrate (pH 6, Sigma-Aldrich) for 30 min in a water bath at 96 °C. Sections were left to cool down to room temperature (RT) for 30 min before washing in 1× Tris buffer saline (TBS). Sections were incubated for 30 min in a blocking buffer containing 5% bovine serum albumin (BSA, Sigma-Aldrich) and 20% normal donkey serum (Neuromics) in TBS. Respective primary antibodies were added to the sections and incubated overnight at 4 °C. After washing in TBS, secondary antibodies were added and incubated for 1 h at RT. All antibodies were diluted in blocking buffer and are listed in Supplementary Data 1C with their respective dilution and antigen retrieval buffer. After washing in TBS, sections were stained with DAPI solution (1:1000 in blocking buffer) (final concentration 1 μg/mL, Thermo Fisher Scientific) for 10 min and washed shortly in TBS. Sections were then mounted with coverslips in Prolong Gold Antifade Reagent (Invitrogen) and imaged within 1 week. Images were acquired using a Nikon ultrafast widefield microscope (Nikon) and processed using FIJI software (version 2.14.0, ImageJ2)[80] and OMERO. To control unspecific binding of the secondary antibodies, negative controls were prepared, as shown in Supplementary Fig. 5C.

### Immunofluorescence staining image analysis

Heterogenous illumination of DDX4 raw images was corrected using FIJI software v2.14.0[80] with the BaSic plug-in[81]. We then used CellProfiler v4.2.5 to create an image analysis pipeline[82]. First, we created a mask for delimitating tissue pieces as a region of interest (ROI) based on Gaussian-filtered and dilated DAPI signals from illumination-corrected images. Secondly, we identified follicles as objects using a distance morph on the inverted DAPI signal. Here, we incorporated a quality control step allowing for manual inspection of follicles (i.e. deleting objects that do not correspond to follicles and adding objects that have been missed by the algorithm). Thirdly, we measured the mean intensity of DDX4 inside the follicles, as well as in the tissue region outside of follicles (labeled as background) for each image. We subtracted background intensity from follicle intensity for each image individually. Data was processed and visualized using R studio.

### Quantitative PCR

To test the effect of follicle division on the oocyte and granulosa marker expression qPCR analysis was performed. Isolated from adult ovaries, 10 primary and 10 secondary follicles were lysed, and cell lysate was divided into two, and both halves of the lysate were used in separate qPCR reactions (Supplementary Data 1A). Due to the small number of cells in the primary and secondary follicles, the starting point was cell lysate without RNA extraction, and cDNA was amplified before qPCR using the smart-seq2 protocol as described in the Smart-seq2 library preparation and sequencing section.

Primers used for qPCR analysis were designed using NCBI primer-blast (https://www.ncbi.nlm.nih.gov/tools/primer-blast/). Preferred

primers amplified all isoforms of the selected gene and spanned exon-exon junction. Primers sequences used are the following: *DDX4* (F' CAAGCCGCGGAGAGAACTTG; R' GATGATGAAGCTGGAGTCCTGTTA), *DAZL* (F' GTTCCAGCGGACCTCACAG; R' TCAGGATTTGCAGTAGA-CATGATGG), *FOXL2* (F' TAAGCTCCTGTCGCTCCTCT; R' CTTTCCG CGGTGAATTTGGG) and housekeeping gene *UBC* (F' TTCCGTCGCAG CCGGG; R' TGCATTGTCAAGTGACGATCACAG).

The qPCR analysis was carried out on LightCycler 480 instrument (Roche). For the detection of mRNA expression Platinum SYBR Green qPCR SuperMix (Invitrogen) was used. Each sample was run in triplicates on a 384-well plate for up to 40 amplification cycles. The specificity of amplified PCR products was controlled using melt curve analysis.

## RNA fluorescent in situ hybridization (RNA FISH)

RNA expression levels of human ovarian follicles were visualized using RNAscope Fluorescent Multiplex kit v2 platform (PN. 323110, advanced cell diagnostics (ACD), and all steps were performed according to the provided protocol (RNAscope Fluorescent Multiplex kit v2 user manual) with some alterations. All incubations not on RT were performed in HybEZ oven (ACD), and all reagents used are part of the RNAscope platform by ACD, unless indicated differently. Briefly, 4 μm thick formalin-fixed paraffin-embedded tissue sections on SuperFrost plus Gold slides (VWR) were baked for 1 hr at 60 °C prior to deparaffinization and rehydration. For better tissue section attachment, slides were then baked again for 30 min at 60 °C. Sections were treated with 5% hydrogen peroxide (PN. 322381) for 20 min at RT to reduce the autofluorescent signal. Target retrieval was then performed by submerging tissue sections in target retrieval solution (PN. 322000) and incubated in a water bath at 96 °C. Sections are then baked again at 60 °C for 30 min to ensure tissue section attachment. Sections were treated with protease plus solution (PN. 322381) prior to probe hybridization and signal amplification. RNAscope probes used in experiments were designed by the manufacturer. The following probes were used to detect mRNA expression: *DDX4* (HS-DDX4-O1-C3, PN. 518571, Channel 3) and *AMH* (HS-AMH-C2, PN. 1086791, Channel 2). Positive control was performed using human specific 3-plex positive control probe targeting housekeeping genes Peptidylprolyl Isomerase B (*PPIB*, Channel 2) and Ubiquitin C (*UBC*, Channel 3) (PN. 320861). Negative control was performed using a 3-plex negative control probe targeting bacterial gene dihydrodipicolinate reductase (*DapB*, all channels, PN. 320871). Channels were then developed, and respective TSA vivid fluorophores were added in a 1:1500 dilution (fluorophores 570 and 650, Tocris Bioscience). Sections were counterstained with DAPI (PN. 323110) and slides were mounted using Prolong Gold Antifade mounting medium (Invitrogen). Images were acquired within 1 week using a Nikon ultra-fast widefield microscope (Nikon) and processed using FIJI software (version 2.14.0, ImageJ2) and OMERO Figure (OME).

## Statistics and reproducibility

Statistical analyses for gene expression were performed using R software version 4.1.1. Gene expression statistics were calculated using the Wald test and adjusted for multiple comparisons with the BH method. Correlations between mRNA and miRNA were calculated using a two-sided Spearman correlation test and adjusted for multiple comparisons using the BH method. Functional enrichment analyses were performed using the online tool (g:Profiler[76] https://biit.cs.ut.ee/gprofiler/gost), and functional enrichment statistics were calculated using Fisher's one-tailed test and adjusted for multiple comparisons using the BH method. Over-representation of miRNAs was analyzed using the online tool (miEAA[79] https://ccb-compute2.cs.uni-saarland.de/mieaa/), calculated using Fisher's exact two-tailed test, and adjusted for multiple comparisons using the BH method. Results were considered significant at an adjusted *p* value < 0.05.

No statistical method was used to predetermine the sample size. Patients for follicle isolation and sequencing were selected based on their age (adult or child). No other selection criteria were applied. Sample selection for immunofluorescence validation was based on the characteristics of the ovarian tissue (tissue samples and sections needed to have ovarian follicles). During data analysis, samples with low sequencing quality (low number of reads) were excluded from downstream analysis. The investigators were not blinded to allocation during experiments and outcome assessment.

### Reporting summary

Further information on research design is available in the Nature Portfolio Reporting Summary linked to this article.

## Data availability

The gene expression data (Smart-seq2 and small RNA) generated in this study have been deposited in the Gene Expression Omnibus under the accession codes: GSE241982, GSE241981, and GSE241983. The raw Smart-seq2 and small RNA-sequencing data collected from Helsinki patients are protected and are not publicly available due to Finnish data privacy laws. RNA-sequencing data collected from Helsinki patients will be made available through the Federated European Genome-phenome Archive (FEGA) and its national node that is operated by CSC IT Center for Science Ltd (https://research.csc.fi/-/fega). The FEGA repository is intended for storage and distribution of sensitive omics data. Deposition of these metadata to the FEGA central archive, deposition of these data to the national FEGA node at CSC, and requests to access the data by external researchers will be applied from the local Data Access Committee, formed by the Helsinki University Hospital and the University of Helsinki. Once the data is made available through the FEGA, external researchers can apply access to it through the Sensitive Data Apply service at CSC. We estimate the time for the first response to be within a month and access to data within six months. For more information contact Timo Tuuri (timo.tuuri@helsinki.fi, University of Helsinki). The processed count matrixes of both Swedish and Helsinki samples are available under the accession code: GSE241984. The supporting data generated in this study are provided in the Supplementary Information and in Source data file. Source data are provided with this paper.

## Code availability

No custom codes or algorithms were used to generate results.

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

## Acknowledgements

We would like to express our gratitude to the clinicians and research nurses who assisted us in patient recruitment and biopsy collection, and to the patients who contributed their invaluable materials for our research. We acknowledge Dr. Valentina Di Nisio, Dr. Paula Peltopuro, and Sanna Honkasalo for their assistance in handling ovarian tissue. Additionally, we appreciate the language editing provided by Nicola Byers. We also thank the FENO and Live Cell Imaging Facility/Nikon center for Excellence at Karolinska Institute, supported by the KI Infrastructure Council. This work was funded by the Estonian Research Council (grants PRG1076 A.S. and PSG608 A.V.M.), EU Horizon 2020 innovation grants (ERIN grant no. EU952516 P.D. and A.S., FREIA grant no 825100 P.D. and A.S.), The Swedish Childhood Cancer Foundation (KP2020-0012 K.J., PR2020-0096 P.D.), the Birgitta and Carl-Axel Rydbeck's Research grant for Pediatric Research (2021-00079 K.J.), the Swedish Research Council VR (2020-02132 P.D.), Sigrid Jusélius foundation K.J., the Finnish Cancer Society K.J., the Finnish Pediatric Research Foundation K.J., as well as Karolinska Institutet grants (PhD funding KID and Consolidator Award P.D.).

## Author contributions

I.R. carried out the experiments, analyzed data, performed sequencing library preparation and data analysis, and wrote the manuscript.

J. Hassan, J. Hao., M.W., M.O., K.K., and T.T. carried out the experiments with ovarian tissue. E.M.L. carried out immunofluorescence quantification. K. Jääger carried out sequencing library preparation. C. Lindskog contributed to immunofluorescence experiments K.P., S.G., C.L., H.V., P.F., J.M., and K.J. coordinated patient recruitment and evaluated patient health. K.J., A.S., A.V.M., and P.D. contributed to essential resources, conceived the experiments, supervised the project, and wrote the manuscript. All authors discussed the results, contributed to the preparation of the paper, and approved the final version. A.V.M. and P.D. authors jointly supervised this work.

## Funding

## Competing interests
The authors declare no competing interests.

## Additional information

[1]Division of Obstetrics and Gynecology, Department of Clinical Science, Intervention and Technology, Karolinska Institutet, Huddinge, Stockholm, Sweden. [2]Department of Gynecology and Reproductive Medicine, Karolinska University Hospital, Stockholm, Sweden. [3]Department of Chemistry and Biotechnology, Tallinn University of Technology, Tallinn, Estonia. [4]Department of Reproductive Medicine, Xiangya Hospital, Central South University, Changsha, PR China. [5]Institute of Computer Science, University of Tartu, Tartu, Estonia. [6]Competence Centre on Health Technologies, Tartu, Estonia. [7]Department of Obstetrics and Gynecology, Helsinki University Hospital, University of Helsinki, Helsinki, Finland. [8]Department of Immunology, Genetics and Pathology, Uppsala University, Uppsala, Sweden. [9]Department of Women's and Children's Health, Karolinska Institutet, Stockholm, Sweden. [10]Department of Pediatric Oncology, Queen Silvia Children's Hospital, Sahlgrenska University Hospital, Gothenburg, Sweden. [11]Crown Princess Victoria Children's Hospital, and Division of Children's and Women's Health, Department of Biomedical and Clinical Sciences, Linköping University, Linköping, Sweden. [12]Department of Women's and Children's Health, Uppsala University Children's Hospital, Uppsala, Sweden. [13]Astrid Lindgren Children's Hospital, Karolinska University Hospital, Stockholm, Sweden. [14]Childhood Cancer Research Unit, Department of Women's and Children's Health, Karolinska Institutet, Stockholm, Sweden. [15]Department of Obstetrics and Gynaecology, Institute of Clinical Medicine, University of Tartu, Tartu, Estonia. [16]Children's Hospital, University of Helsinki and Helsinki University Hospital, Helsinki, Finland. [17]Department of Women's and Children's Health, NORDFERTIL Research Lab Stockholm, Karolinska Institutet and University Hospital, Stockholm, Sweden. [18]These authors jointly supervised this work: Agne Velthut-Meikas, Pauliina Damdimopoulou. ✉e-mail: ilmatar.rooda@ki.se; pauliina.damdimopoulou@ki.se

