## [Peer Review File · Nature Communications]

REVIEWER COMMENTS

Reviewer #1 (Remarks to the Author):

This manuscript written by Rooda et al. determined and analyzed transcriptomes in ovarian cortical follicles from child and adult human. With in-depth analysis of the transcriptomes, the authors found an alternative type of follicles that is clearly distinct from canonical follicles. The alternative follicles are represented by lack of oocyte-marker gene expression, enhanced expression of granulosa cells, and potential secretion of signaling molecules. The authors identified differentially expressed genes (DEGs) by comparison of transcriptomes obtained from different sets of follicles: adult vs child, resting vs activated primordial follicles, and with vs without chemotherapy. Based on the comparison, a prominent difference between adult and child follicles was observed in gene expression program involved in theca cell function at a secondary follicle stage. The first signature during follicular activation is represented by genes including WT1, FOXL2, YAP and FGF2. Effect of chemotherapy with the cyclophosphamide equivalent dose (CED) on gene expression was dose-dependent and represented by genes related to interferon signaling, but not to DNA-damage response. CED does not affect the potential function of the alternative type of follicles.

This manuscript provides valuable and comprehensive transcriptomic information on human child and adult ovarian follicles. The authors performed a noteworthy and well-organized method, capturing images of the follicles prior to RNA extraction. This approach allows for a retrospective inspection of the relationship between morphology and transcriptomic data. For this reason, this manuscript makes a unique contribution to the understanding of follicular development in the human ovary, notwithstanding the fact that various groups, including the author's, have previously published single-cell analyses of human ovaries. Especially, an alternative type of follicles, which potentially provide signals for other follicles, is quite intriguing. Furthermore, the identification of DEGs during follicular activation will serve crucial molecular markers to enhance our understanding of early follicular development, which remains largely unexplored. Due to its valuable resources and unique aspects, this manuscript is recommended for publication in Nature Communications. However, there are several points that need clarification for further improvement, outlined as follows:

1. Looking at the list of the materials, Patient 7 is the only one who provide “non-treated” child ovary. While I acknowledge the challenges in obtaining such a sample, there is a slight concern that relying on only one patient may introduce bias into the results. Can the author mitigate this possibility by conducting an evaluation of the dataset (Patient 7) through some methods? Alternatively, adding another dataset (although challenging) could help address this concern.

2. The individual patient dots in Figure F2I are hard to distinguish. Could the author consider displaying separate violin plots for each patient? In addition, it would be informative to show more representative immunofluorescence images of Group1- and Group2-follicles in a supplementary figure.

3. The circle plot and chord plot illustrating the interaction between Group1- and Group2-follicles (Figure 2J and SU Figure 3A) are intriguing. Two comments on these plots: Could the author provide a brief explanation to help readers easily follow the visualizations? What if the author illustrated the effect of Group1 on Group2 using a chord plot, mirroring the approach taken in Supplementary Figure 3A (the effect of Group2 on Group1)?

4. L423: We notice heterogeneity in the transcriptomes of morphologically similar follicles, even within the same group (Group1). This is confusing, because the author previously described that Group1 include follicles various developmental stages.

5. If possible, it would be beneficial if the authors could display all follicular images captured prior to RNA extraction and establish a link to the corresponding transcriptomic data.

Reviewer #2 (Remarks to the Author):

Manuscript by Rooda et al presents detailed transcriptomic analysis of human follicles collected from cryopreserved ovarian cortex of either children (prior onset of puberty) or adults. Authors set to compare how age of patients impacts transcriptomic profile single follicles from primordial, primary and secondary stages, representing pooled RNA from both granulosa cells and oocytes. They create a gene patterns based on follicle stage and patient age and use various bioinformatics tools to mine this data for understanding of human follicle development.

Authors also explore changes with treatment to chemotherapy. As ovarian tissue has been obtained from donors with previous treatments (chemotherapy for children and androgen exposure prior sex reassignment surgery). Human ovaries, especially from young age are difficult to obtain and thus caveats of treatment, if properly acknowledged, are unavoidable as control unexposed tissues would be close to impossible to obtain. What manuscript is lacking and what could be accomplished is acute treatment (with and without exposure to chosen drug) of cortical strips in vitro followed up by transcriptional analysis of follicles. This would help to establish the immediate vs delayed transcriptional responses in a controlled environment. I would strongly encourage authors to consider this approach.

The most interesting and thought-provoking concept of the manuscript is a possibility of existence of two follicular fates, where functional outcome of one set of follicles is to create an offspring (oocytes with full developmental competence) and the other would-be endocrine role for regulation of female reproductive axis. While this concept has been previously suggested in animals (especially mouse), human data to support this are lacking. However, to fully support their hypothesis, authors should properly analyse the data to correct for stage. If authors wish to claim that there are 2 kinds of follicles, stage appropriate comparison is necessary. From their current analysis it is not clear if oocyte DEGs observed are driven by stage of development or represent true transcriptional differences at given stage. From numbers in Fig 1 it should be possible to have sufficient sample size for analysis of individual follicles by stage (especially primary and secondary), to determine if proposed hypothesis is correct. Also, perhaps pooling data from validation set (additional 46 follicles in transitional stages mostly) with original set may also boost the power of analysis.

The same criticism applies to quantitation of fluorescence presented in Fig 2H. Stage appropriate comparison should be made.

Authors should clarify if validation set of data (qPCR and smart seq2) came from independent set of ovaries or was done from the same patient cohort, just different strip of cortex.

Reviewer #3 (Remarks to the Author):

The manuscript by Rooda et al. is the first time that early follicles from adult and child ovaries are molecularly analyzed. Overall, the manuscript will be very valuable to the field of reproductive biology and is written well, the experiments are technically, and the data are bioinformatically processed well. There are only a few places where the manuscript can be improved:

1. Because of the wealth of data, a lot of information is incorporated into the figures. It is therefore very important that the font size in all of the figures and the quality of the images are reasonable for publication. For example, the histology and other images in Figure 1 are not high enough quality. Figure 4D and F panels are other extreme examples where data figures and font size are minuscule.
2. Likewise, the font size in the supplemental tables are extremely small.
3. Although the authors mention the androgen treatment in the 4 adult women (lines 468-472), androgens are still critical in the ovary in their conversion to estrogens, and it is still possible that short term androgen treatment may have some effect on the follicles and the data generated. The discussion should therefore mention this and the authors should not bypass this important issue.

4. In lines 407-409, the authors reference the second statement but fail to reference the first point. The authors should add Dong et al. Nature 1996 because this is the first example where TGF-beta signaling pathways were shown to be essential for early ovarian folliculogenesis.

Reviewer #4 (Remarks to the Author):

Rooda et al set to perform an unbiased transcriptomic comparison between cortical follicles from children and adults, in the process the authors discover that, unrelated to the child-adult comparison, follicles split into two groups. The child-adult comparison is timely and important, but the comparison is limited by: 1) small sample size (4 adults versus 5 children) and 2) confounding by treatment (all adults were treated with androgen and 4 out of 5 children were treated with chemotherapy, both treatments have the potential to affect follicles in different ways). The unexpected observation that follicles from both adults and children separate into two groups is potentially interesting and warrants further analyses.

Major points:

1) I find the DE analysis between group 1 and 2 in Figure 2 uninterpretable because it confounds the interesting phenotype (that some follicles in the same stage separate into two groups) with an uninteresting one (that group 1 has a large number of primordial follicles while group 2 does not, I say this is "uninteresting" because DE between primordial and other follicle stages has been described before and that's not the strength nor major selling point of this paper). Connected to this, I do not find the summary in lines 244-249 and 525-574 convincing: group 2 follicles might have lower expression of essential oocyte markers because the signal is diluted due to these follicles having more granulosa and theca cells than follicles in group 1. For this claim it would be essential to perform the same DE analysis between group 1 and group 2 but stratified by follicle stage (e.g. group 1 primary versus group 2 primary). This would be a much more interpretable analysis.

2) Related to my comment above: AMH is used as a marker for group 2 in SU Figure 2C, but it is clear from main Figure 2C that AMH is expressed in all primary and secondary follicles regardless of whether they are in group 1 or group 2, again suggesting that the group 1 versus group 2 DE analysis may contain true signals but is confounded by follicle stage.

3) I have the concern that the analyses presented have used follicle as the unit of replication without statistically controlling for individual. Follicles from the same individual cannot be considered to be independent from one another, therefore it would be important to model individual effect, for example by using individual as a random effect in a mixed model. I would guess that many of the results presented will not be significant when individual is taken into account and this uncertainty should be acknowledged in the results.

4) In the follicle-follicle communication analysis using CellChat (Fig. 2J) what exact sample normalisation method was used? I could not find this specified in the methods. I am concerned that a simple normalisation of dividing by library size may have been used. This would not be appropriate as it does not account for differences in RNA composition between follicles. I expect that a difference in RNA composition exists between primordial, primary and secondary follicles due to differences in oocyte size and number of granulosa cells per follicle. I would suggest that the authors show that using alternative normalization methods, like DESeq2's median of ratios or EdgeR's trimmed mean of M values (TMM), does not change the results.

5) The number of significant DE genes (statistical power) is dependent on sample size. Since the number of follicles in each stage is likely not the same, the numbers of DEGs in Figure 3B, 3G and 4A are not directly comparable. The authors would need to run their DE analysis with subsampling and permutation to make them so.

6) Absence of evidence is not evidence of absence. In the analyses in section 2.6 the sample sizes are very very small so I'm not surprised that the authors do not find effects, for example in line 471 the length of androgen therapy did not have major effects on follicle gene expression, yet the sample size is only 4 (even if there are multiple follicles per individual - see my comment #2). A careless reader will read that there is no effect of androgen. This result should be much more tempered or even replaced with "sample size was too small to discern if androgen therapy has an effect".

7) There are 6 instances of "data is not shown" in this manuscript. I think it is important to substantiate all claims with data.

8) It would be important to specify the number of individuals in which validations were performed in the main text (e.g. line 190).

Minor comments:

- Figure 2I: I got confused by the placement of the "Group 1" and "Group 2" labels on this panel. I would suggest the author put this label below and above the red line instead of left and right. The font size of the colour key needs to be increased, I cannot read it.

- Figure 2J: I find this panel very uninformative, I cannot see from it the pattern described in the text. I would suggest the authors replace it with SU Figure 3A.

- The introduction would benefit from a motivation for looking at miRNAs in the context of follicle growth.

- I would suggest that the authors use the same nomenclature for Group 1 and 2 in the text and discussion instead of introducing a new Type 1 and Type 2.

Reviewer #1 (Remarks to the Author):

This manuscript written by Rooda et al. determined and analyzed transcriptomes in ovarian cortical follicles from child and adult human. With in-depth analysis of the transcriptomes, the authors found an alternative type of follicles that is clearly distinct from canonical follicles. The alternative follicles are represented by lack of oocyte-marker gene expression, enhanced expression of granulosa cells, and potential secretion of signaling molecules. The authors identified differentially expressed genes (DEGs) by comparison of transcriptomes obtained from different sets of follicles: adult vs child, resting vs activated primordial follicles, and with vs without chemotherapy. Based on the comparison, a prominent difference between adult and child follicles was observed in gene expression program involved in theca cell function at a secondary follicle stage. The first signature during follicular activation is represented by genes including WT1, FOXL2, YAP and FGF2. Effect of chemotherapy with the cyclophosphamide equivalent dose (CED) on gene expression was dose-dependent and represented by genes related to interferon signaling, but not to DNA-damage response. CED does not affect the potential function of the alternative type of follicles.

This manuscript provides valuable and comprehensive transcriptomic information on human child and adult ovarian follicles. The authors performed a noteworthy and well-organized method, capturing images of the follicles prior to RNA extraction. This approach allows for a retrospective inspection of the relationship between morphology and transcriptomic data. For this reason, this manuscript makes a unique contribution to the understanding of follicular development in the human ovary, notwithstanding the fact that various groups, including the author's, have previously published single-cell analyses of human ovaries. Especially, an alternative type of follicles, which potentially provide signals for other follicles, is quite intriguing. Furthermore, the identification of DEGs during follicular activation will serve crucial molecular markers to enhance our understanding of early follicular development, which remains largely unexplored. Due to its valuable resources and unique aspects, this manuscript is recommended for publication in Nature Communications. However, there are several points that need clarification for further improvement, outlined as follows:

1. Looking at the list of the materials, Patient 7 is the only one who provide “non-treated” child ovary. While I acknowledge the challenges in obtaining such a sample, there is a slight concern that relying on only one patient may introduce bias into the results. Can the author mitigate this possibility by conducting an evaluation of the dataset (Patient 7) through some methods? Alternatively, adding another dataset (although challenging) could help address this concern.

Response:

We appreciate the reviewer for highlighting this concern, which is very familiar to us as well. In our analyses, all comparisons between adult and child follicles were conducted using all child samples, both untreated and treated, collectively.

Child follicles were divided into groups when chemotherapy exposure effect was analyzed. This evaluation was first performed in two distinct groups: one clinically not considered as

having ovarian damage (cumulative CED < 6 000 g/m², patients 5 and 7), and another clinically considered as having an increased risk of ovarian damage (cumulative CED ≥ 6,000 g/m², patients 6, 8, and 9). This comparison is relevant from a clinical fertility preservation perspective and also provides greater statistical reliability with a more comparable number of patients in each category. Then, to further increase reliability, we estimated the impact of CED dose on the expression level of the top DEGs, revealing significant dose-response relationships. These data are in Figure 6.

Importantly, the validation set samples were derived from a different set of children, including a non-exposed patient (patient 27, Supplementary Table 1). This independent set was used when the RNA-seq findings were validated on the protein level by immunostainings. This has been clarified in the manuscript (lines 493-494 and 541-542).

Finally, we considered if the differences between adult and child follicles might be related to the fact that some of the child follicles had been exposed to chemotherapy. To investigate this, we compared the DEGs related to chemotherapy in children (n=166 between the no risk of ovarian damage group and the elevated risk of ovarian damage group) to the DEGs related to age group (n=261 between child and adult). We observed a low overlap, with only 8 genes found in both comparisons (*IFI44L*, *ISG15*, *IFIT1*, *DDIT4*, *NWD1*, *CNTN2*, *LINC01206*, and *AL450405.1*). Therefore, the differences detected between adult and child follicles are likely attributed to age effects, with chemotherapy exerting some influence but not significantly affecting these outcomes.

2. The individual patient dots in Figure F2I are hard to distinguish. Could the author consider displaying separate violin plots for each patient? In addition, it would be informative to show more representative immunofluorescence images of Group1- and Group2-follicles in a supplementary figure.

Response:

We thank the reviewer for pointing out this shortcoming. We have updated the panel to include three separate violin plots (Figure 2I). In addition, we have replaced the immunofluorescent images (Supplementary Figure 2D) with figures that display a larger number of follicles in the frame.

3. The circle plot and chord plot illustrating the interaction between Group1- and Group2-follicles (Figure 2J and SU Figure 3A) are intriguing. Two comments on these plots: Could the author provide a brief explanation to help readers easily follow the visualizations? What if the author illustrated the effect of Group1 on Group2 using a chord plot, mirroring the approach taken in Supplementary Figure 3A (the effect of Group2 on Group1)?

Response:

We appreciate the reviewer's feedback. In response to the suggestion, as well as comments from other reviewers, we have made adjustments to the main and supplementary figures concerning follicle communications. We have removed Figure 2J and replaced it with SU Figure 3A to improve clarity. Additionally, we have enhanced the legend for Figure 2J with a more

detailed explanation. We hope that these changes will make it easier for the reader to follow the results.

Furthermore, we have carried out an analysis of the communication from Group 1 to Group 2 and included the resulting circle plot in Supplementary Figure 3B. These new findings indicate that Group 1 follicles also secrete outgoing signals that Group 2 follicles can potentially receive. However, Group 1 follicles demonstrate a lower probability of communication compared to Group 2 primary follicles, aligning with the updated Figure 2J, which illustrates the overall communicator/receiver strengths among the different follicles. Notably, the top predicted signalling molecules secreted by both Group 2 and Group 1 follicles are similar and include signals such as MK, SEMA3, GAS, and AMH.

Although we did not originally include communications from Group 1 to Group 2 follicles in the manuscript, we recognize the importance of highlighting this and have thus incorporated these results into Supplementary Figure 3 and adjusted the results section accordingly, page 10, lines 234-237.

“Although Group 1 follicles also exhibited the capability to secrete a range of growth factors, including both previously identified ones and novel factors, the predicted signaling strength was notably lower compared to Group 2 follicles (Figure 2J and SU Figure 3B).”

4. L423: We notice heterogeneity in the transcriptomes of morphologically similar follicles, even within the same group (Group1). This is confusing, because the author previously described that Group1 include follicles various developmental stages.

Response:

We thank the reviewer for bringing this matter to our attention. In the sentence in question, we discuss the heterogeneity among follicles of the same stage, specifically primordial follicles, which all belong to Group 1. We agree that the original phrasing of this sentence was misleading. We have now made corrections to enhance the clarity and readability. Page 21, lines 536-537.

“We noticed heterogeneity in the transcriptomes of same stage follicles, even when they all belonged to the same group (e.g. Group 1).”

5. If possible, it would be beneficial if the authors could display all follicular images captured prior to RNA extraction and establish a link to the corresponding transcriptomic data.

Response:

We appreciate the reviewer’s input on this matter. In response, we have added images of all follicles involved in the study to Supplementary Figure 2A. Each individual follicle is labelled in accordance with the same principles used in the count matrix, which is available under the accession number GSE241984. We hope that this connection between morphology and gene expression offers valuable insights into follicular dynamics.

Reviewer #2 (Remarks to the Author):

Manuscript by Rooda et al presents detailed transcriptomic analysis of human follicles collected from cryopreserved ovarian cortex of either children (prior onset of puberty) or adults. Authors set to compare how age of patients impacts transcriptomic profile single follicles from primordial, primary and secondary stages, representing pooled RNA from both granulosa cells and oocytes. They create a gene patterns based on follicle stage and patient age and use various bioinformatics tools to mine this data for understanding of human follicle development.

Authors also explore changes with treatment to chemotherapy. As ovarian tissue has been obtained from donors with previous treatments (chemotherapy for children and androgen exposure prior sex reassignment surgery). Human ovaries, especially from young age are difficult to obtain and thus caveats of treatment, if properly acknowledged, are unavoidable as control unexposed tissues would be close to impossible to obtain.

1. What manuscript is lacking and what could be accomplished is acute treatment (with and without exposure to chosen drug) of cortical strips *in vitro* followed up by transcriptional analysis of follicles. This would help to establish the immediate vs delayed transcriptional responses in a controlled environment. I would strongly encourage authors to consider this approach.

Response:

We appreciate the reviewer's understanding of the general caveats of working with human ovarian tissue, which are indeed all too familiar to us. Access to such tissue is limited and strictly regulated, and the sample size is always very small. We have been fortunate to collaborate with various clinics to assemble the study presented in the manuscript, and we have made every effort to ensure the validity of the findings reported.

Indeed, the idea of controlled *in vitro* exposures is highly valid, and our laboratory has a long history of working with cortical strip cultures. Over the years, we have learned to understand the model better, including its possibilities and limitations. Traditionally, the impact of culture/exposure on ovarian tissue has been assessed histologically by visual analysis of follicles in hematoxylin-eosin-stained tissue sections. These analyses have often been complemented by targeted analysis of markers, such as immunostained apoptosis/DNA damage markers (see for example PMID: 36481098 and PMID: 36723906). As we have established robust transcriptomic methods in our group, we have started assessing cortical tissue in culture using RNA-sequencing, which provides a more holistic and unbiased overview of gene expression changes on a global level. What we have found is that the mere culture of ovarian cortex significantly influences gene expression, altering the expression of thousands of genes within a 24-hour culture period, with many being associated with inflammatory processes (*e.g.*, TNF α signaling via NF- κ B, interferon response), (please see Rebuttal Letter Figure 1A-D, adapted from Li *et al* in revision). Our manuscript describing these results is under revision for Human Reproduction Open (Li *et al.* in revision).

These studies have made it painfully evident that simple cortex culture, especially with short exposure, is predominantly affected by tissue damage responses. It is exceedingly challenging to demonstrate the impacts of chemical exposures on gene expression atop this general

damage response. For this reason, we, alongside many others, have begun to develop alternative models to mimic human ovaries in culture, such as organoids. Unfortunately, these models are not yet ready to be applied in research on toxicity.

An alternative to *in vitro* cortical strip culture could be mouse xenograft models. Numerous previous studies have employed xenograft models to investigate the effects of chemotherapy on human ovarian follicles¹⁻⁵. These studies have consistently demonstrated that chemotherapy can activate DNA damage and apoptosis pathways, leading to the loss of primordial follicles. Notably, the effect of chemotherapy on DNA damage becomes apparent in the xenografted human ovarian tissue as early as 12 hours after exposing the mouse to chemotherapy¹. Therefore, the immediate response within the ovarian cortex is likely to primarily involve DNA damage response and apoptosis. Our tissue samples were harvested from patients at much later time points, and the responses we observed could be long-term side effects.

In summary, we are currently unable to conduct the suggested experiment with the available models. However, we are working very hard to develop alternative models that we hope will enable us to explore questions related to mechanisms of damage in human ovaries in the future. Nevertheless, we have adjusted the results section by incorporating additional literature information about the effect of chemotherapy exposure on ovarian follicles, and added further details about the chemotherapy timing in our patient samples (page 24, lines 545-553).

“Previous experiments with adult and fetal ovarian tissue xenograft models have shown that exposure to alkylating chemotherapy induced DNA damage response in ovarian follicles [44] and a reduction in primordial follicle densities as early as 12-24 hours after exposure [45]. The lack of DNA damage response markers in our dataset could depend on time since exposure. In our study, ovarian tissue was collected 1-44 days after the last chemotherapy dose, while the time since the first chemotherapy exposure varied from 26 to 113 weeks. These time frames are long, and the initial DNA damage response could have already disappeared. Instead, our data suggest upregulated interferon signalling as a longer-term side-effect of chemotherapy in follicles.”

Rebuttal Letter Figure 1. Overview of transcriptomic changes after 24 hours of ovarian cortex culture with or without *in-vitro* activation of follicles. Human ovarian cortical tissue was cut into fragments and cultured for 24 hours in the presence of *in vitro* activators (PTEN inhibitor and PI3K activator, “Frag + IVA”) or without activators (“Frag”). Tissues were then collected for bulk RNA-sequencing along with fresh uncultured samples. A) Principal component analysis of the RNA-sequencing data showing strong segregation of the samples to fresh and cultured. B) Number of differentially expressed genes between the different groups. C) Clustered heatmap of average expression of inflammatory response-related genes at 24 h in the RNA-sequencing data. D) Barplot showing selected significantly enriched hallmark gene sets from GSEA analysis using significant differentially expressed genes in Frag vs fresh and Frag + IVA vs fresh comparisons (FDR < 0.1). Normalized enrichment score (NES). These data are in revision to Human Reproduction Open (Li *et al.* in revision).

Rebuttal Letter references:

- Oktem O, Oktay K. A novel ovarian xenografting model to characterize the impact of chemotherapy agents on human primordial follicle reserve. *Cancer Res.* 2007;67(21):10159-10162. doi:10.1158/0008-5472.CAN-07-2042

2. Soleimani R, Heytens E, Darzynkiewicz Z, Oktay K. Mechanisms of chemotherapy-induced human ovarian aging: double strand DNA breaks and microvascular compromise. *Aging (Albany NY)*. 2011;3(8):782-793. doi:10.18632/aging.100363
3. Li F, Turan V, Lierman S, Cuvelier C, De Sutter P, Oktay K. Sphingosine-1-phosphate prevents chemotherapy-induced human primordial follicle death. *Hum Reprod*. 2014;29(1):107-113. doi:10.1093/humrep/det391
4. Meng Y, Xu Z, Wu F, et al. Sphingosine-1-phosphate suppresses cyclophosphamide induced follicle apoptosis in human fetal ovarian xenografts in nude mice. *Fertil Steril*. 2014;102(3):871-877.e3. doi:10.1016/j.fertnstert.2014.05.040
5. Wu M, Xue L, Chen Y, et al. Inhibition of checkpoint kinase prevents human oocyte apoptosis induced by chemotherapy and allows enhanced tumour chemotherapeutic efficacy. *Hum Reprod*. 2023;38(9):1769-1783. doi:10.1093/humrep/dead145

2.The most interesting and thought-provoking concept of the manuscript is a possibility of existence of two follicular fates, where functional outcome of one set of follicles is to create an offspring (oocytes with full developmental competence) and the other would-be endocrine role for regulation of female reproductive axis. While this concept has been previously suggested in animals (especially mouse), human data to support this are lacking. However, to fully support their hypothesis, authors should properly analyse the data to correct for stage. If authors wish to claim that there are 2 kinds of follicles, stage appropriate comparison is necessary. From their current analysis it is not clear if oocyte DEGs observed are driven by stage of development or represent true transcriptional differences at given stage. From numbers in Fig 1 it should be possible to have sufficient sample size for analysis of individual follicles by stage (especially primary and secondary), to determine if proposed hypothesis is correct. Also, perhaps pooling data from validation set (additional 46 follicles in transitional stages mostly) with original set may also boost the power of analysis.

Response:

We appreciate the reviewer for highlighting this apparent concern when mixing varying proportions of follicular stages in differential gene expression analysis. We acknowledge the importance of conducting equivalent stage follicle comparisons to mitigate the potential impact of differing developmental stages on the reported outcomes. Such comparisons, specifically Group 1 primary versus Group 2 primary, and Group 1 secondary versus Group 2 secondary, were indeed performed, but we admit that the results were not explicitly included in the manuscript. Considering the importance of this point, the brief one-sentence summary we had incorporated was not sufficient.

To address this concern more comprehensively, we have heeded the reviewer's suggestion and included the detailed functional enrichment analysis results in Supplementary Table 2. In addition, we provide a plot illustrating the top enriched processes in Group 1 and Group 2 primary follicles in Supplementary Figure 1J. These results stem from the DEGs obtained through the equivalent stage comparison. We believe that this addition enhances the clarity

and depth of our analysis. We hope that this adjustment addresses the reviewer's concerns and contributes to the overall robustness of our study. Page 9, lines 199-203.

“We also considered that the results from the GO analysis might be skewed due to the unequal distribution of follicle stages between Group 1 and Group 2. However, when we compared Group 1 and Group 2 follicles of the same developmental stage, similar results were obtained (SU Table 2C-F and SU Figure 1J).”

3.The same criticism applies to quantitation of fluorescence presented in Fig 2H. Stage appropriate comparison should be made.

Response:

We thank the reviewer for highlighting this issue. The classification of pre-antral cortical follicles relies on factors such as shape, size, color, and the number of granulosa cell layers surrounding the oocyte. As human ovarian tissue is susceptible to fixation-related artifacts, Bouin fixation followed by hematoxylin-eosin staining is the preferred method for preserving tissue morphology for accurate follicle staging. However, Bouin's fixative is harsh and not well suited for antibody stainings. The analysis mentioned in Figure 2H employs formalin-fixed tissue and immunofluorescent staining, which makes the assessment of follicle stage based on morphology very challenging. Although granulosa cells can be identified by their nuclear DAPI staining, this nuclear stain does not reveal cell shape, making the organization of cells difficult to interpret. As a result, classifying follicles based solely on the DAPI signal is unreliable. For this reason, our quantification includes all cortical follicles, where we know that the majority are in the primordial stage. To mitigate the lack of stage-specific information, we have quantified a significant number of follicles (n=343).

4.Authors should clarify if validation set of data (qPCR and smart seq2) came from independent set of ovaries or was done from the same patient cohort, just different strip of cortex.

Response:

We thank the reviewer for pointing this unclarity out. qPCR, test sequencing, and validation patients were different from the main study patients presented in Table 1. The number of patients and follicles used for those analyses is highlighted in Supplementary Table 1A-C. We made changes to the text to improve the readability. Page 8, lines 159-165.

“Therefore, we carried out qPCR analysis of selected genes in both halves of split follicles from a different set of patients (n=2, GRP patients, n=20 follicles). We found that the results correlated significantly (SU Figure 1C, SU Table 1A SU Methods) suggesting that the splitting yields two homogeneous samples. We also sequenced an additional independent validation set comprising 46 follicles isolated from an independent set of patients (n=4, three GRP and one child fertility preservation patient), without splitting them (SU Table 1B, SU Methods).”

Reviewer #3 (Remarks to the Author):

The manuscript by Rooda et al. is the first time that early follicles from adult and child

ovaries are molecularly analyzed. Overall, the manuscript will be very valuable to the field of reproductive biology and is written well, the experiments are technically, and the data are bioinformatically processed well. There are only a few places where the manuscript can be improved:

1. Because of the wealth of data, a lot of information is incorporated into the figures. It is therefore very important that the font size in all of the figures and the quality of the images are reasonable for publication. For example, the histology and other images in Figure 1 are not high enough quality. Figure 4D and F panels are other extreme examples where data figures and font size are minuscule.

Response:

We appreciate the reviewer for highlighting this shortcoming. Indeed, the quality of Figure 1 had dropped significantly in the version automatically compiled by the submission system for peer review. We have addressed this issue by adjusting the histology images in Figure 1 and saving the figure in a different format to prevent a decline in quality in the future. Additionally, we reviewed all figures and adjusted font sizes to enhance readability. We hope that these modifications contribute to improved clarity and readability of the images.

2. Likewise, the font size in the supplemental tables are extremely small.

Response:

We thank the reviewer for bringing this matter to our attention. We have increased the font size in all supplementary tables.

3. Although the authors mention the androgen treatment in the 4 adult women (lines 468-472), androgens are still critical in the ovary in their conversion to estrogens, and it is still possible that short term androgen treatment may have some effect on the follicles and the data generated. The discussion should therefore mention this and the authors should not bypass this important issue.

Response:

We thank the reviewer for pointing out this shortcoming. We have rewritten this part of the results section and hopefully have improved the readability of the results. Page 23, lines 591-596.

“The length of androgen therapy did not have major effects on follicle gene expression. This is in line with our previous study, where single-cell profiles of ovarian cortical tissue derived from c-sec patients and GRPs did not differ markedly [21]. In addition, ~30% of GRPs on testosterone show signs of ovulation [41]. Nevertheless, it is imperative to acknowledge the potential existence of a nuanced androgenic impact on pre-antral follicles. Our study’s limited sample size may have hindered our ability to discern such subtleties.”

4. In lines 407-409, the authors reference the second statement but fail to reference the first point. The authors should add Dong et al. Nature 1996 because this is the first example where TGF-beta signaling pathways were shown to be essential for early ovarian folliculogenesis.

Response:

We thank the reviewer for pointing this out. A suggested reference was added to the sentence.

Reviewer #4 (Remarks to the Author):

Rooda et al set to perform an unbiased transcriptomic comparison between cortical follicles from children and adults, in the process the authors discover that, unrelated to the child-adult comparison, follicles split into two groups. The child-adult comparison is timely and important, but the comparison is limited by: 1) small sample size (4 adults versus 5 children) and 2) confounding by treatment (all adults were treated with androgen and 4 out of 5 children were treated with chemotherapy, both treatments have the potential to affect follicles in different ways). The unexpected observation that follicles from both adults and children separate into two groups is potentially interesting and warrants further analyses.

Major points:

1) I find the DE analysis between group 1 and 2 in Figure 2 uninterpretable because it confounds the interesting phenotype (that some follicles in the same stage separate into two groups) with an uninteresting one (that group 1 has a large number of primordial follicles while group 2 does not, I say this is "uninteresting" because DE between primordial and other follicle stages has been described before and that's not the strength nor major selling point of this paper). Connected to this, I do not find the summary in lines 244-249 and 525-574 convincing: group 2 follicles might have lower expression of essential oocyte markers because the signal is diluted due to these follicles having more granulosa and theca cells than follicles in group 1. For this claim it would be essential to perform the same DE analysis between group 1 and group 2 but stratified by follicle stage (e.g. group 1 primary versus group 2 primary). This would be a much more interpretable analysis.

Response:

We appreciate the reviewer for highlighting this concern, which Reviewer 2 also mentioned. We recognize the importance of conducting equivalent stage follicle comparisons to minimize the potential impact of stage variations on follicle grouping. Although these comparisons (Group 1 primary versus Group 2 primary and Group 1 secondary versus Group 2 secondary follicles) were performed, we realize the results were not explicitly stated in the manuscript but only summarized as a brief sentence.

In response to this feedback, we have included details of the DEGs from equivalent stage comparisons between Group 1 and Group 2 follicles in the revised manuscript. These findings are now detailed in the supplementary information (Supplementary Table 2). Furthermore, we have conducted enrichment analyses and present the results in Supplementary Figure 1J. We believe that this addition improves the clarity and depth of our analysis. We hope that these revisions adequately address the reviewer's concerns and enhance the robustness of our study. Page 9, lines 199-203.

"We also considered that the results from the GO analysis might be skewed due to the unequal distribution of follicle stages between Group 1 and Group 2. However, when we compared

Group 1 and Group 2 follicles of the same developmental stage, similar results were obtained (SU Table 2C-F and SU Figure 1J)."

2) Related to my comment above: AMH is used as a marker for group 2 in SU Figure 2C, but it is clear from main Figure 2C that AMH is expressed in all primary and secondary follicles regardless of whether they are in group 1 or group 2, again suggesting that the group 1 versus group 2 DE analysis may contain true signals but is confounded by follicle stage.

Response:

We thank the reviewer for highlighting this point. Indeed, *AMH* is expressed in both Group 1 and Group 2 follicles. We agree that it is difficult to illustrate nuances in expression levels through *in situ* staining techniques, such as that presented in Supplementary Figure 2C, where *AMH* was used as a marker to identify Group 2 follicles, which express low levels of *DDX4* and high levels of *AMH*. We believe that it is apparent in the image that the expression level of *AMH* (indicated by the yellow signal) varies among follicles, although categorizing signal intensity as "high" or "low" precisely is challenging. To enhance clarity, we revised the description in the legend of Supplementary Figure 2C. Rather than suggesting a definite distinction between high and low expressing follicles, we now describe variations in expression levels as "higher" and "lower" to reflect the differences more accurately (page 10, lines 216-219).

"Furthermore, RNA in situ hybridization experiments validated the presence of follicles exhibiting varying levels of DDX4 and AMH transcripts within the ovarian cortex both in children and adults (SU Figure 2C)."

3) I have the concern that the analyses presented have used follicle as the unit of replication without statistically controlling for individual. Follicles from the same individual cannot be considered to be independent from one another, therefore it would be important to model individual effect, for example by using individual as a random effect in a mixed model. I would guess that many of the results presented will not be significant when individual is taken into account and this uncertainty should be acknowledged in the results.

Response:

We appreciate the reviewer for bringing this issue to our attention. In response, we re-ran the comparisons in Figure 3B, this time controlling for the patient factor. This adjustment resulted in a reduced number of differentially expressed (DE) genes in two of the comparisons, while for seven other comparisons, it yielded a higher number of DE genes between follicle stages (Rebuttal Letter Figure 2A). It is important to note that excluding the patient factor from the calculation introduces more variability into the statistical model, leading to more conservative test statistics and, consequently, a reduced number of DE genes compared to when the patient is included.

Moreover, we examined the overlap between adult and child gene expression changes during follicle growth using the corrected model, similar to our approach in the non-corrected analysis shown in Figure 3C. The results showed a similar percentage of common genes, along

with a higher proportion of adult-specific genes and a lower proportion of child-specific genes (Rebuttal Letter Figure 2B).

We also assessed the overlap between the two models (with and without considering individual patients) and found that the majority of genes were detected in both models (Rebuttal Letter Figure 2C-D). Furthermore, we conducted pattern analysis and GO enrichment analysis following the same principle as in Figure 3D-F with the patient-corrected results. The enrichment results showed an overlap in the top 10 enriched GO terms between both analyses (Rebuttal Letter Figure 2E-F, with terms identical in both corrected and non-corrected results marked in red).

Despite the significant overlap of DEGs between the models, some genes were specific to a particular model. Thus, we agree with the reviewer's observation regarding the level of uncertainty in the results, which may be influenced by individual variation. Consequently, we have added sentences to the results section to highlight this point. Page 13, lines 350-353. In addition, we have included the alternative model results in Supplementary Figure 4A and added both gene lists to the "Source Data" file.

"The observed differential expression of genes in the dataset may be subject to some degree of uncertainty due to variations among patients and the limited sample size. However, even when the statistical model was controlled for the patient, similar results were obtained (SU Figure 4A)."

Rebuttal Letter Figure 2. Comparison of patient-corrected and non-corrected models. A) Comparative analysis of differentially expressed genes (DEGs) across follicular stages, without controlling for the patient and with controlling for the patient (in brackets). B) Overlap of adult and child DEGs in the patient-corrected model. C) Comparison of adult DEGs between the two models. D) Comparison of child DEGs between the two models. E) Enriched GO terms associated with upregulated gene patterns found in the patient-corrected models (children and adult). The terms that are identical to those in the non-corrected model, presented in Figure 3E, are highlighted in red. F) Enriched GO terms associated with downregulated gene pattern (children and adults) found in the patient-corrected models. The terms that are identical to those in the non-corrected model, presented in Figure 3E, are highlighted in red.

4) In the follicle-follicle communication analysis using CellChat (Fig. 2J) what exact sample normalisation method was used? I could not find this specified in the methods. I am concerned that a simple normalisation of dividing by library size may have been used. This would not be appropriate as it does not account for differences in RNA composition between follicles. I expect that a difference in RNA composition exists between primordial, primary and secondary follicles due to differences in oocyte size and number of granulosa cells per follicle. I would suggest that the authors show that using alternative normalization methods, like

DESeq2's median of ratios or EdgeR's trimmed mean of M values (TMM), does not change the results.

Response:

We appreciate the reviewer for raising this concern. In our analysis, samples were initially normalized using the CellChat normalization function (normalizeData), which incorporates library-size normalization with a scale factor of 10,000, followed by log-transformation with a pseudocount of 1.

In response to the suggestion, we conducted the same analysis using normalized counts from DESeq2 and EdgeR. Interestingly, across all three normalization methods, Group 2 follicles consistently exhibited higher outgoing interaction strength and Group 1 primordial and intermediary follicles displayed very low incoming and outgoing signal strength (Rebuttal Letter Figure 3). Furthermore, significant pathways from Group 2 to Group 1 showed overlap among the tested methods, including pathways such as SEMA3, GAS, KIT and AMH.

To enhance readability, we updated the methods section to include the normalization information. Page 38, lines 1010-1012.

"In brief, the loaded count matrix was normalized using normalizeData function (scale factor 10 000) and log transformed with a pseudocount of 1."

Rebuttal Letter Figure 3. Scatter plot overview of cell-to-cell communications among follicles. A) CellChat normalization; B) Deseq2 normalization and C) EdgeR normalization.

5) The number of significant DE genes (statistical power) is dependent on sample size. Since the number of follicles in each stage is likely not the same, the numbers of DEGs in Figure 3B, 3G and 4A are not directly comparable. The authors would need to run their DE analysis with subsampling and permutation to make them so.

Response:

We thank the reviewer for highlighting this matter. In designing the study, our goal was to make the age groups and follicle stage groups as similar as possible. While the number of ovarian follicles in each stage and across the two age groups is comparable, they are not identical (Summarized in Figure 1, detailed in Supplementary Table 3). To evaluate the

robustness of our differential expression (DE) analysis results, we followed the reviewer's recommendation and conducted a bootstrapping analysis. This involved generating 200 bootstrap samples from our original dataset through random sampling with replacement. Subsequently, DE analysis using DESeq2 was performed on each bootstrap sample to identify DE genes. To determine the significance of our DE results, we compared them with those from permuted datasets, which were created by randomly shuffling the group labels to assess the probability of identifying the DE genes by chance.

The comparison between adult and child follicles (Figure 4A) revealed a high frequency of DE genes in both the DE subsampling and permutation analysis. For different follicle stages, the average frequency of DE genes was notable: 86.75 (range 45-110) for primordial follicles, 91.13 (range 53-107) for intermediate follicles, 86.42 (range 61-108) for primary follicles, and 98.18 (range 51-130) for secondary follicles, suggesting the DE genes identified are unlikely due to chance. Additionally, bootstrapping analyses for primordial-to-primary and primary-to-secondary stage-wise comparisons (Figure 3B) for child follicles revealed average frequencies of DE genes at 98.56 (range 27-135) and 85.79 (range 34-113), respectively. Similarly, for adult follicles, the average frequency of DE genes was 99.83 (range 59-137) and 93.42 (range 40-126), respectively. Overall, these findings indicate that the DE genes, despite varying follicle counts across comparisons, are unlikely to be incidental.

We included the frequency values of DE genes between adult and child follicles in Supplementary Table 6A-D.

6) Absence of evidence is not evidence of absence. In the analyses in section 2.6 the sample sizes are very very small so I'm not surprised that the authors do not find effects, for example in line 471 the length of androgen therapy did not have major effects on follicle gene expression, yet the sample size is only 4 (even if there are multiple follicles per individual - see my comment #2). A careless reader will read that there is no effect of androgen. This result should be much more tempered or even replaced with "sample size was too small to discern if androgen therapy has an effect".

Response:

We thank the reviewer for pointing out this shortcoming, which was also noted by Reviewer #3. The recommended change was applied. Page 23, lines 591-596.

"The length of androgen therapy did not have major effects on follicle gene expression. This is in line with our previous study, where single-cell profiles of ovarian cortical tissue derived from c-sec patients and GRPs did not differ markedly [21]. In addition, ~30% of GRPs on testosterone show signs of ovulation [42]. Nevertheless, it is imperative to acknowledge the potential existence of a nuanced androgenic impact on pre-antral follicles. Our study's limited sample size may have hindered our ability to discern such subtleties."

7) There are 6 instances of "data is not shown" in this manuscript. I think it is important to substantiate all claims with data.

Response:

We thank the reviewer for bringing our attention to this issue. We have added the missing information to the supplementary information. Supplementary Table 2C-F now includes details on the functional enrichment analysis for Group 1 and Group 2 primary and secondary follicles. Additionally, we have created an extra Supplementary Figure 7, showing a principal component analysis of microRNAs and CED exposure (Supplementary Figure 7A), the expression of DNA damage response genes following CED exposure (Supplementary Figure 7B), and a comparison of interferon pathway gene expression in Group 1 and Group 2 follicles (Supplementary Figure 7C).

8) It would be important to specify the number of individuals in which validations were performed in the main text (e.g. line 190).

Response:

We thank the reviewer for bringing this matter to our attention. The number of individuals in each validation set was added to the text. In addition, Supplementary Table 1A-C contains information on the validation patients.

Minor comments:

- Figure 2I: I got confused by the placement of the "Group 1" and "Group 2" labels on this panel. I would suggest the author put this label below and above the red line instead of left and right. The font size of the colour key needs to be increased, I cannot read it.

Response:

We thank the reviewer for pointing this matter to our attention. We have changed the location of "Group 1" and "Group 2" as suggested under and above the red line and we have modified the font size of the legend.

- Figure 2J: I find this panel very uninformative, I cannot see from it the pattern described in the text. I would suggest the authors replace it with SU Figure 3A.

Response:

We thank the reviewer for this suggestion. We have removed Figure 2J and replaced it with the Supplementary Figure 3A panel.

- The introduction would benefit from a motivation for looking at miRNAs in the context of follicle growth.

Response:

We thank the reviewer for this suggestion. A short introduction to the miRNAs was added. Page 4, lines 78-81.

"MicroRNAs (miRNAs) are short non-coding RNAs expressed in ovarian somatic cells, oocytes, and follicular fluid. While miRNAs have been extensively studied in large pre-ovulatory follicles, there is limited knowledge about their expression and role in human pre-antral follicles [6]."

- I would suggest that the authors use the same nomenclature for Group 1 and 2 in the text and discussion instead of introducing a new Type 1 and Type 2.

Response:

We thank the reviewer for pointing this out. The change in nomenclature was intended to provide a more biologically relevant designation for the two groups of follicles, as the names Group 1 and 2 merely referred to the two visually distinct clusters of follicles in the UMAP in Figure 1B. We thought that referring to them as Type 1 and Type 2 would highlight their biological differences, potentially encouraging other researchers to adopt these terms if studying similar phenomena. However, if this alteration causes confusion among readers, we will maintain consistent nomenclature throughout the manuscript. We have changed all Type 1 and Type 2 back to Group 1 and Group 2.

REVIEWER COMMENTS

Reviewer #1 (Remarks to the Author):

In the revised manuscript, the authors have addressed all concerns raised in the first round of review. The revised manuscript has become more informative and, therefore, is valuable for publication in Nature Communications.

Reviewer #3 (Remarks to the Author):

This is an outstanding manuscript. The authors have done an excellent job to improve the manuscript in this second submission. My critiques were addressed appropriately and adequately.

Reviewer #4 (Remarks to the Author):

I thank the authors for their reviews and hope that they have found my comments useful. I still have the following outstanding comments:

1) The authors did not fully address my previous major comment nr 1. My concrete recommendation still is that the relevant parts of Figure 2 (D-G and accompanying text) are replaced by the stratified analysis results rather than by the current version (perhaps showing only one stage in the main figure and the rest in supplement), which is feasible since the authors already performed the analysis. This would improve the clarity of the analysis.

2) It is important the authors provide a full list of DEGs for each analysis (as far as I could tell, the lists in Supplementary Table 2 show only high-level gene set enrichment results).

3) Similarly, the new patient-controlled results should replace main Figure 3 and the list of DEGs should be provided in supplement. How was the patient effect modelled? This seems to be missing from the methods.

4) The analysis of cell-cell communication using different normalisation methods is appreciated, but not included in the manuscript. I recommend that the authors include this as supplementary material.

5) I apologise if my comment regarding power in Figure 3B, G, etc was not quite clear. These panels show numbers of differential expressed genes, these numbers are not comparable to one another if the number of follicles in each stage is not the same (e.g. the comparison between child primordial and intermediate follicles has a total sample size of 27 (8 vs 19), whereas the comparison between child intermediate and primary has a total sample size of 46). Hence my suggestion of using downsampling so that each comparison uses the same number of samples. It is important that these panels show the number of DEGs from the downsampling analysis, possibly with ranges from permutations. It is possible that the authors have done this but the downsampling is not described and the main figures do not appear to have been changed in the manuscript to reflect it.

6) The definition of child and adult-specific genes in Figure 3C-E and 3H-I is not statistically sound. Please see: <https://www.ncbi.nlm.nih.gov/pmc/articles/PMC9046926/> for recommendations on modelling an interaction effect instead of overlapping list of genes.

Reviewer #4 (Remarks to the Author):

I thank the authors for their reviews and hope that they have found my comments useful. I still have the following outstanding comments:

1) The authors did not fully address my previous major comment nr 1. My concrete recommendation still is that the relevant parts of Figure 2 (D-G and accompanying text) are replaced by the stratified analysis results rather than by the current version (perhaps showing only one stage in the main figure and the rest in supplement), which is feasible since the authors already performed the analysis. This would improve the clarify of the analysis.

Response: We appreciate the reviewer for highlighting this concern. Following the suggestion, we have updated Figure 2 to include a comparison between Group 1 primary and Group 2 primary results. Additionally, the results of secondary follicles are available in Supplementary Figure 1.

2) It is important the authors provide a full list of DEGs for each analysis (as far as I could tell, the lists in Supplementary Table 2 show only high-level gene set enrichment results).

Response: We appreciate the reviewer for bringing up this point. We have made adjustments to enhance the clarity and accessibility of our data. Indeed, Supplementary Table 2 contains enrichment results based on genes with a log₂ fold change greater than 1.5. We understand the value of providing the full lists of DEGs in Supplementary information for readers' convenience. As such, we have updated Supplementary Table 2A-F to include functional enrichment results, while Supplementary Table 2G-I now contains the DEG lists from the three comparisons (primary follicle groups, secondary follicle groups, and all follicles in group comparisons).

3) Similarly, the new patient-controlled results should replace main Figure 3 and the list of DEGs should be provided in supplement. How was the patient effect modelled? This seems to be missing from the methods.

Response: We thank the reviewer for this comment. To address this concern, we have made adjustments to the presentation of our results. Specifically, we have replaced non-patient-controlled results in Figure 3 with patient-controlled results and removed non-controlled results from the manuscript. Furthermore, to enhance clarity, we have relocated the lists of patient-controlled DEGs from the "Source data" file to Supplementary Table 4.

We recognize the importance of transparency in methodology and apologize for the oversight in not including a description of how patient correction was performed in the revised manuscript methods (Page 39, lines 1016-1018).

4) The analysis of cell-cell communication using different normalisation methods is appreciated, but not included in the manuscript. I recommend that the authors include this as supplementary material.

Response: We thank the reviewer for this suggestion. We added a comparison of three normalization methods to Supplementary Figure 3C-E.

5) I apologise if my comment regarding power in Figure 3B, G, etc was not quite clear. These panels show numbers of differentially expressed genes, these numbers are not comparable to one another if the number of follicles in each stage is not the same (e.g. the comparison between child primordial and intermediate follicles has a total sample size of 27 (8 vs 19), whereas the comparison between child intermediate and primary has a total sample size of 46). Hence my suggestion of using downsampling so that each comparison uses the same number of samples. It is important that these panels show the number of DEGs from the downsampling analysis, possibly with ranges from permutations. It is possible that the authors have done this but the downsampling is not described and the main figures do not appear to have been changed in the manuscript to reflect it.

Response: We thank the reviewer for the clarification. Figure 3 results are obtained using only Group 1 follicles and therefore the numbers are smaller than the reviewer suggests. Nevertheless, we have carried out down-sampling as suggested.

Down-sampling to an equal number of follicles across all stages resulted in 6 follicles for adult comparisons and 4 follicles for child comparisons. We performed 5 sets of down-sampling with a random selection of follicles and compared the overlap with the full set of follicle results. In adult samples, the full set captured 80-96% of genes present in down-sampling results. However, in child samples, with fewer follicles, the full set captured 20-76% of genes. These comparisons are summarized in SU Figure 4C.

Given that down-sampling is not an ideal strategy for such a limited number of follicles, we decided to keep the full dataset of patient-controlled results in Figure 3, as the reviewer recommends in comment 3. Additionally, a down-sampling summary table has been included in Supplementary Figure 4C, and lists of DEGs from all sets and follicle comparisons are available in the Source data file. Furthermore, we addressed the issue of uncertainty in our results, particularly regarding the variability in follicle numbers, in lines 348-351 "The observed differential expression of genes in the dataset may be subject to some degree of uncertainty, for instance, due to inter-patient variability, limited sample size, and unequal number of follicles between different comparisons."

6) The definition of child and adult-specific genes in Figure 3C-E and 3H-I is not

statistically sound. Please

see: <https://www.ncbi.nlm.nih.gov/pmc/articles/PMC9046926/> for recommendations on modelling an interaction effect instead of overlapping list of genes.

Response: We thank the reviewer for bringing this to our attention. We acknowledge the limitations of using a Venn diagram as the primary method for gene selection in downstream analysis. However, this is not what we did in our manuscript. The Venn diagram, together with the pattern analyses, was used specifically to illustrate that DEGs found in one age group behaved the same in the other age group. As the reviewer points out, focusing solely on DEGs would mask a lot of information. We emphasize that the pattern analysis was carried out using all DEGs. We have now removed the "age group specific" and "common" split of the patterns. Instead, we show adult DEGs and child DEGs patterns and explain that the overall changes are the same, highlighted by the other age-group being displayed in the background. The downstream GO analyses were carried out on the patterns that behave the same in children and adults, not on age-group specific DEGs.

We hope that the new way of displaying the results is clearer to the readers. We emphasize that the adjustment of the figure does not change any of the results, as the previous version of the pattern analysis was done on all DEGs too. We hope that this new approach to show the results ensures a comprehensive approach that considers the entirety of the data without selective bias and without misleading the reader.

REVIEWERS' COMMENTS

Reviewer #4 (Remarks to the Author):

The authors have now addressed all of my concerns.